# Sexual inhibition and sexual excitation in a sample of Polish women

**Krzysztof Nowosielski** [1,2]☯*, **Jacek Kurpisz**[3]☯, **Robert Kowalczyk**[4]☯

**1** Faculty of Medical Sciences, Department of Gynecology and Obstetrics, Medical University of Silesia, Katowice, Poland, **2** Institute of Medical Sciences, University of Opole, Opole, Poland, **3** Department and Clinic of Psychiatry, Pomeranian Medical University, Szczecin, Poland, **4** Department of Sexology, Andrzej Frycz Modrzewski Krakow University, Cracow, Poland

☯ These authors contributed equally to this work.
* gabiner@drnowosielski.pl, knowosielski@sum.edu.pl

**Data Availability Statement:** All files are available for the Mendeley database - http://dx.doi.org/10.17632/t3pnbj2g66.1.

**Funding:** The authors received no specific funding for this work.

## Abstract

The contemporary concept of sexual counseling for women with sexual problems, distress, and female sexual dysfunction (FSD) includes tailored medical and/or psychological intervention. The dual control model and the Sexual Excitation/Sexual Inhibition Inventory for Women (SESII-W) are helpful for identifying risk factors and tailoring therapy for FSD. The current study aimed to (1) validate the Polish translation of the SESII-W in a sample of Polish women, and (2) verify the usefulness of the SESII-W in clinical practice. Five hundred nine white women age 18 to 55 years old ($M \pm SD$ age = 39.7 ± 11.3 years) were included in this cross-sectional study. Linguistic validation of the Polish translation of the SESII-W was first performed. A battery of tests was then used to evaluate reliability, convergent and discriminant validity, measurement invariances, and correlations between the SESII-W and other measures. Given that the original version of the SESII-W had unsatisfactory model fit, exploratory and confirmatory factor analyses were subsequently performed. Results showed a new final model that included 26 items with seven lower- and two higher-order factors and explained 58.9% of the variance in the data, with CFI = 0.93, RMSEA = 0.05 and $\chi^2$ = 693.39, $p < 0.001$. Cronbach's α was 0.77 for Sexual Excitation (SE) and 0.88 for Sexual Inhibition (SI) scales. A moderate negative association between SI and the presence of FSD according to Diagnostic and Statistical Manual of Mental Disorders, 5th Edition (DSM-5) criteria was noted. SE was positively associated with engaging in risky sexual behaviors, Extraversion and Openness to Experiences traits, and was negatively correlated with relationship quality. Finally, age was negatively correlated with all domains of the SESII-W except Arousal Contingency. SE and SI were both lower in older women as compared to younger once. These results demonstrate that the Polish version of SESII-W shows good psychometric properties. A higher propensity for SI was associated with the presence of sexual problems, distress, and FSD, whereas a higher propensity for SE was associated with greater engagement in risky sexual behaviors and personality type. However, future studies on larger and more diverse populations are required to confirm the replicability of the factor structure of the scale.

**Competing interests:** The authors have declared that no competing interests exist.

## Introduction

The reason to engage or not in variety of sexual behaviors is regulated on different levels. One of the mechanisms that regulate sexual response is based on dual control model (DCM) theory. The basis of the DCM theory was presented by Jansen and Bancroft in 1999, who postulated the existence of two systems–inhibitory and excitatory [1]. At the trait level, the inhibitory and excitatory sexual systems are independent, but at the state level, they mutually moderate behavior. Both systems play a role in regulating sexual desire, wherein a high propensity for sexual inhibition may serve as a major factor in the etiology of sexual dysfunctions and a high propensity for sexual excitation may lead to engaging in risky sexual behaviors (RSB) [1–3].

The inhibition and excitation sexual systems should be considered in the broader context of Gray's theory of behavioral inhibition and excitation systems that relate to dopaminergic and serotoninergic pathways in the brain, particularly in the limbic system [4]. Previous studies suggest that sexual inhibition and excitation is, at least in part, independent of the general behavioral control system in the brain. Furthermore, personality traits described by Eysenck, including extraversion (i.e., positive emotionality), neuroticism (i.e., negative emotionality), and psychoticism (i.e., constraint) alongside the dimension of anxiety and impulsivity, might contribute to the activation/inhibition system, also on the level of sexual control. Sexual excitation/inhibition might also be seen in the broader concept of risk avoidance/risk taking, when activity of inhibition system is high and low, respectively. DCM also assumes that the individual propensity for sexual inhibition/excitation results in either engaging or restricting from uncommitted and multiple sexual relationships, sexual risk taking, and different patterns in response to sexual cues (i.e., disgust vs. enjoyment and entertainment) [3,5].

To measure the aforementioned propensities, a sexual excitation/inhibition scales were developed for both men and women: the Sexual Inhibition (SI) and Sexual Excitation (SE) Scale–SIS/SES for men and women [5,6], the Sexual Excitation/Sexual Inhibition Inventory for Women–SESII-W [7], and the Sexual Excitation/Sexual Inhibition Inventory for Women and Men–SESII-W/M [8]. Short forms of the SESII-W have been recently developed [9].

The present study focused on the SESII-W scale, given that it is one of the most widely used scales to measure sexual inhibition/excitation. We also selected the SESII-W because it was specifically designed for women and enables more in-depth analysis of different dimensions of female sexual function. The SESII-W is a self-reported questionnaire that measures the propensity for sexual excitation and sexual inhibition. The original scale [7] consists of 36 question with 4 possible answers that range from 1 (strongly disagree) to 4 (strongly agree). The scale has 8 subscales or domains that correspond to the following eight lower-order factors: (1) Arousability, (2) Partner Characteristics, (3) Sexual Power Dynamics, (4) Smell, (5) Setting (Unusual or Unconcealed), (6) Concerns about Sexual Function, (7) Arousal Contingency, and (8) Relationship Importance. Subscales 1–5 constitute the Sexual Excitation (SE) higher-order factor whereas subscales 6–8 constitute the Sexual Inhibition (SI) factor. The scale has been shown to have good internal consistency (except for Concerns about Sexual Function and SI; Cronbach's α = 0.63 and 0.55, respectively), with the highest Cronbach's α observed for Arousability (α = 0.80). Further, test-retest reliability of the original scale ranges from 0.51 (Arousal Contingency) to 0.82 (Sexual Inhibition).

Validation studies for the German [10], Dutch [11], and Spanish [12] translations of the SESII-W have been recently published. However, these studies noted some differences in the measure as compared to the original validation study.

In particular, the original structure of the SESI-WW questionnaire was retained only in the German validation study [10], with 36 items grouped in eight lower-order factors and two higher-order factors. However, in that study, some items had low loading onto factors and

some cross-loadings were observed [10]. Further, the model presented in the German validation study had unsatisfactory model fit, CFI = 0.85. Despite these limitations, the German SESII-W scale demonstrated adequate internal consistency, except for Partner Characteristics and Sexual Power Dynamics subscales (Cronbach's α = 0.58 and 0.46, respectively). All factors demonstrated good to excellent test-retest reliability (Pearson's r between 0.66 and 0.83).

Both the Dutch [11] and Spanish translations of the SESII-W [13] modified the original structure of the scale. The Dutch version included 35 items grouped in eight subscales and two higher-order factors, as in the original study. The Spanish version included 33 items with two higher-order factors; however, the structure of this scale differed from the original scale (i.e., Setting was moved from Sexual Excitation and included in Sexual Inhibition). The Dutch model demonstrated unsatisfactory fit indices (CFI = 0.87). While the Spanish model demonstrated satisfactory fit (RMSEA = 0.04, CFI = 0.92, TLI = 0.91), some cross-loadings between subscales were observed. For example, Sexual Power Dynamics and Concerns about Sexual Function loaded onto both SE and SI factors. Based on these three prior validation studies, we attempted to be stricter while analyzing the data to allow for items to load onto different lower order factors than in the original scale. Our goal was to obtain the highest model fit while minimizing cross-loadings.

Further validation studies on DCM are necessary, given the growing number of studies showing its utility in clinical practice. For example, results of Clifton et al. suggest that SE (measured by SESII-W) affects both genital and subjective arousal (Partner Characteristics and Settings positively, Sexual Power Dynamics–negatively), whereas SI affects only the subjective, more conscious components of arousal [14]. A study by Velten et al. confirmed that Setting positively predicts subjective arousal, whereas Concerns about Sexual Function negatively predicts objective arousal [15]. Unpublished data by Bradford show that the Arousal Contingency domain is positively correlated with trait anxiety, as measured by the State-Trait Anxiety Inventory (STAI) (r = 0.38) [16]. Similar results were obtained in a study by Carpenter et al., who used the SIS/SES scale for women and found that state anxiety (as measured by the STAI) was correlated with SI but not with other domains [6]. Based on the confirmed bidirectional relationship between depression and/or the presence of depressive symptoms and sexual function [17], Velten et al. showed that Arousability, Sexual Power Dynamics, Setting, Concern about Sexual Function, Arousal Contingency and depression predict sexual functioning [18]. Although the effects of depression on the inhibitory/excitatory system are not well established, lower levels of serotoninergic associated with depression and higher levels of catecholamine associated with anxiety may increase overall inhibitory tone while having only minor effects on excitatory tone [16]. A few recent studies [18–28] showed a correlation between sexual functioning and SE/SI, such that women with sexual problems, sexual distress, or other sexual concerns had lower SE scores (primarily in Arousability) and higher SI scores as compared to women without sexual concerns. However, the aforementioned studies used screening questionnaires, such as the Female Sexual Function Index (FSFI) and Female Sexual Distress Scale (FSDS), to assess sexual problems. To our knowledge, only one study conducted in the Netherlands [11] used DSM-IV criteria to assess sexual dysfunction. Additionally, existing validation studies on the SESII-W [10–12,26] or other SE/SI measurements [18,22,24] showed some discrepancies in scale structure and cultural and/or gender differences. Additionally, a study by Dang et al. [29] demonstrated ethnic differences in SE between white and Chinese undergraduate students in universities in Canada.

Given the temporal stability of SE/SI [30] and the utility of DCM in clinical practice [11,14,15], further studies in different cultural backgrounds and with current diagnostic criteria for sexual dysfunction are needed.

The present study aimed to fulfill that need for several reasons.

Firstly, it focuses on a cultural adaptation of SESII-W in a Central/Eastern European society. The new version of the SESII-W is expected to put the concept of DCM in the perspective of Eastern European countries, which will allow to examine the DCM from a new, less sexually liberal, perspective. The new Eastern European perceptive can be attributed to two main forces that shaped the concept of sexuality in Poland and other post-soviet countries: socialism and the Catholic Church [31–33]. Given that conservative Catholicism was seen as the answer to socialistic tendencies, Catholicism shaped the contemporary restrictive model of sexual attitudes and behaviors in Polish society. Although many changes have been introduced, some of those restrictive tendencies are still at play and can alter the perception of excitatory cues and inhibitory factors. Given that religion may affect sexual behaviors and attitudes [34], the new version of the SESII-W may better capture individual differences in the propensity for excitation/inhibition in Central and Eastern European countries because it incorporates historical contexts and potential religious influences while retaining the general concept of the DCM. Therefore, the present validation study may prompt future studies in Central and Eastern Europe that may improve understanding of the DCM in a way that is consistent with historical background and conservative religious influences.

Secondly, it uses Diagnostic and Statistical Manual of Mental Disorders, 5th Edition (DSM-5) criteria for sexual dysfunction to verify the correlation between DCM and female sexual dysfunction (FSD).

And finally, it characterizes the propensity for SE and SI in a large sample of women between 18 and 55 years of age.

For all that reasons the results of this study will increase the knowledge on DCM and help to improve the quality of care by introducing personalized sexual medicine [35] for couples with sexual problems

Based on previous research on the DCM and SESII-W [7,10–12,36], we hypothesize that:

1. The Polish model of the SESII-W will differ from the original model, as observed in prior studies. We predict that the Polish version of the SESII-W will require some modification due to sociocultural differences.

2. There will be some degree of overlap between the SESII-W and the following measures: general propensity towards SE/SI, personality, sociosexual orientation, sexual adventurism, and sexual risk taking.

3. Some questions in the Sexual Power Dynamics and Setting subscales concern sexual cues that may be perceived as socially inappropriate, e.g., "Having sex in a different setting than usual is a real turn on for me" or "It turns me on if my partner 'talks dirty' to me during sex". We expect that social norms might influence responses to those questions, given the tendency to respond in socially desirable manner.

4. A higher level of SI will be associated with the presence of sexual distress, sexual problems, sexual dysfunction, depression, and anxiety

5. A higher level of SE will be associated with a greater frequency of masturbation, higher number of lifetime sexual partners, a tendency to engage in RSB, and a better relationship quality and satisfaction.

6. SE will be negatively associated with age, relationship status, religious commitment, and religiosity, whereas positively with sexual activity. The directions are expected to be opposite for SE.

7. The SESII-W will be measurement invariant across different groups (e.g., partnered vs. singles; higher vs. lower education; younger vs. older, online version vs. paper-pencil version), as seen in previous studies [36].

## Methods

### Participants

Nine hundred and seventy-nine white women (18–55 years) were eligible for this cross-sectional population-based study. The inclusion criteria were (1) age between 18 and 55 years and (2) agreement to participate in the study. The respondents were recruited between January 2017 and December 2018 using online social media advertisements (Facebook, local portals) or in outpatient gynecology clinic in Katowice, Poland, during routine yearly check-up visits. The questionnaire was prepared as an online application (posted on surveymonkey.com), as well as a printed paper-pencil version. Both forms (online, traditional printed) can be used in the future, enabling the clinicians to choose the most suitable form. On average, participants completed the study questionnaires in 40 minutes. The paper-pencil form of the questionnaires we as administered to participants during regular clinic visits, completed at home, and then returned (in person or by mail) to the office.

Out of the 979 eligible women, 674 were invited to take part in the online survey and 305 were handed the paper-pencil version of the survey. All participants read an informed consent statement and agreed to participate either verbally (paper-pencil version) or by clicking "YES" in the online survey. Forty-five women refused to participate in the study (11 from the online sample and 34 from the paper-pencil one). Another 374 and 51 respondents returned incomplete online and paper-pencil surveys, respectively. The final sample consisted of 509 participants with complete data (220 paper-pencil surveys and 289 online surveys). Out of the final 509 participants, 42 women from each group participated in the test-retest study which involved completing the questionnaire again after four weeks (see Fig 1). To identify participants for the test-retest procedure, an identification code was distributed. The response rate for the study was 54.5%. The minimum required sample size to perform Confirmatory Factorial Analysis (CFA) was found to be 498, calculated using a Monte Carlo approach [37].

The study protocol was approved by the Ethical Committee of the Silesian Chamber of Physicians and Dentists in Katowice, Poland (ŚIL/KB/756p/15).

### Linguistic validation

The linguistic validation was performed according to 5 step test adaptation procedure recommended by Beaton et al. [38].

In the first step, we compared the translations (i.e., discrepancy resolution), which revealed some discrepancies in items 10, 13, 23, and 30. In item 10, "make me" was translated to "cause me to be"; in item 13, the word "really" was considered to be redundant; in item 23, "can be a turn on" was modified to "can turn me on". In item 30, the word "certain" was also considered redundant. In the next steps, no major inconsistencies were noted, and the first version of SESII-W-PL (Polish version of the scale) was created. Finally, a field test was performed in the group of 25 female students from different departments in the University of Silesia. Difficulties in understanding or interpreting the scale items were recorded, which revealed some deviations in items 11 and 16. Based on these discrepancies, the experts changed "someone" to "partner" in item 11, and "relationship potential" to "material for partner" in item 16. The final version of the SESII-W-PL (Polish version of the scale) was then created.

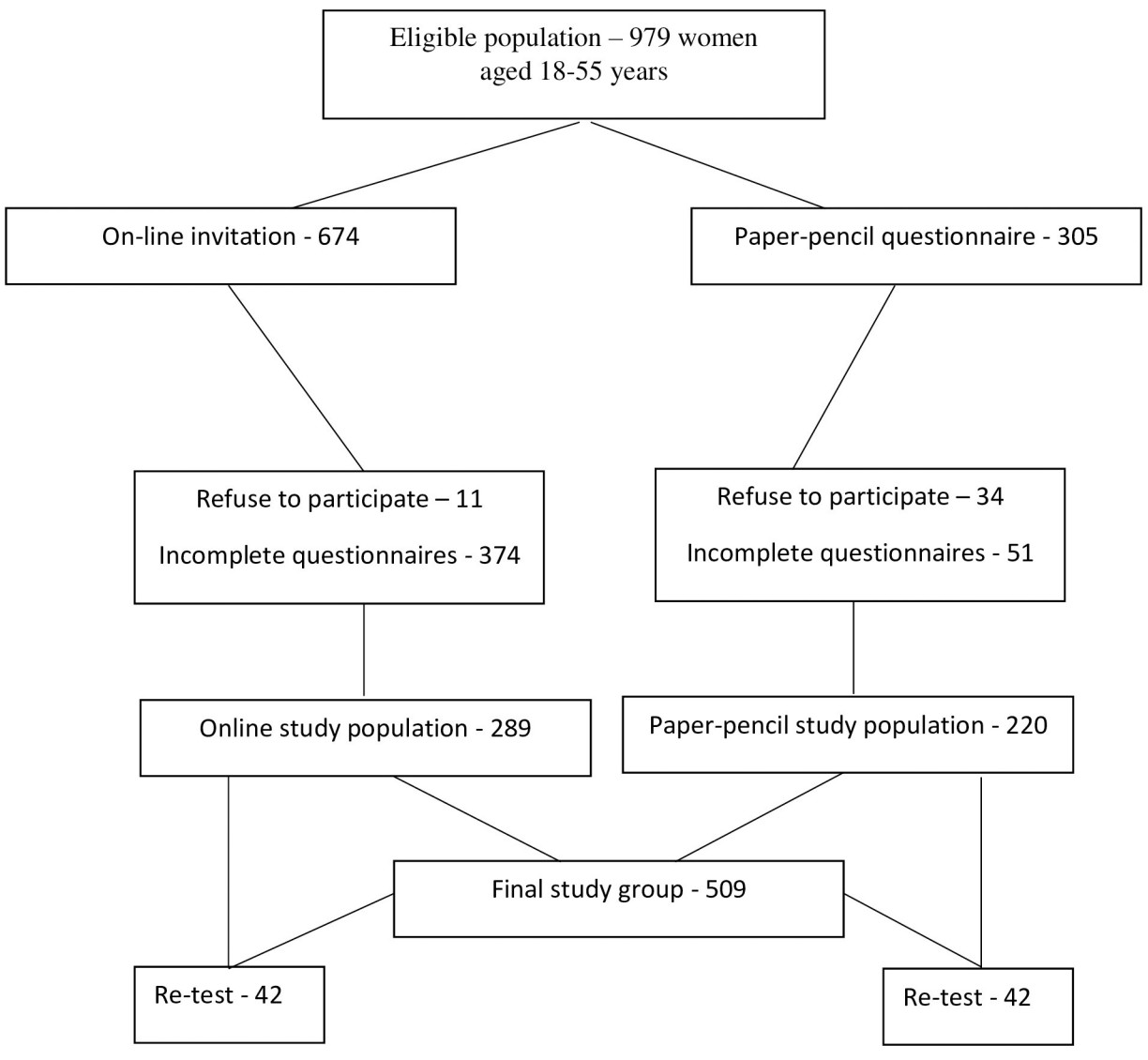

**Fig 1. Study chart: Enrollment protocol.**

## Materials

The survey included questions regarding socioeconomic status, medical history, and sexual behaviors. Participants were also asked to report on their weight and height to calculate body mass index (BMI).

The Hospital Anxiety and Depression Scale (HADS) was used to measure symptoms of depression and anxiety. HADS scores < 10 indicate a lack of depressive symptoms and/or anxiety symptoms whereas scores ≥ 11 points indicate a high risk of clinical depression and/or high levels of anxiety [39]. The HADS was previously validated in Poland and shown a good reliability (Cronbach's α = 0.88) [39].

DMS-5 criteria [40] were used to test for FSD. For the purpose of this study, we used five questions regarding sexual function in the context of DSM-5: (1) "During the last 6 months, how often did you feel like having sex, had sexual fantasies, felt sexually excited and/or felt lubricated in response to sexual stimuli (touch, voice, when seeing a partner, smell, erotic

materials)?"; (2) "During the last 6 months, how often during sexual activities did you experience orgasm or felt sexually fulfilled, that was satisfying enough in the context of time of occurrence, duration, or intensity?"; (3) "During the last 6 months, how often have you been satisfied with your sexual life (emotionally and physically)?"; (4) "During the last 6 months, how often have you experienced problems with penetration (inserting penis to your vagina), anxiety, fear, or unpleasant increased pelvic muscle tension when thinking of, during, or after sexual intercourse?"; (5) "If you were happy with your sexual activity (questions 1–3) during less than 25% of sexual contacts, or if you felt pain or/and anxiety when thinking of, during, or after sex recurrently and permanently in sexual contacts, did it cause distress?" For questions 1–4, the possible answers were as follows: (a) No sexual activity, (b) During less than 25% of sexual contacts, (c) During 25%-50% of sexual contacts, (d) During half of sexual contacts, (e) During 50–75% of sexual contacts, (f) During more that 75% of sexual contacts. For question 5, participants responded either "Yes" or "No". These responses were used for FSD diagnosis according to DSM-5 criteria in both version of the survey. Female Sexual Interest/Arousal Disorder (FSIAD) was diagnosed if sexual interest and/or arousal was present in less than 25% of sexual activities what was associated with sexual distress. Female Orgasm Disorder (FOD) was diagnosed if orgasm and/or satisfaction was experienced in less than 25% of sexual encounters causing distress. Genito-Pelvic Pain/Penetration Disorder (GPPPD) was diagnosed if pain and/or anxiety was present recurrently and permanently during sexual contact what was associated with sexual distress.

The Polish versions of the Female Sexual Function Index (FSFI) and the Female Sexual Distress Scale-Revised (FSDS-R) were used to assess sexual functioning and the presence of sexual problems (FSFS) and sexual distress (FSDS-R), respectively. Higher FSFI scores reflect better sexual functioning whereas higher FSDS scores reflect a higher level of sexual distress. Additionally, women with FSFI scores $\leq$ 27.55 and FSDS-R $\geq$ 13 were classified as being "at risk" of FSD, i.e., distressing sexual concerns. This threshold was based on published recommendations for cut-offs for the population of Polish women [41,42]. Both scales have been validated in Poland and have been shown to have excellent reliability (Cronbach's $\alpha$ = 0.96 and 0.86 for FSFI-PL and FSDS-R, respectively) [41,42].

Respondents were asked to report on their number of lifetime sexual partners, frequency of masturbation ("How often did you masturbate during last the 4 weeks?"), number of any sexual activities in the last 4 weeks (defined as any of the following: vaginal, anal and oral contacts, caressing/cuddling, sexual foreplay, and masturbation), satisfaction from a partner as a lover (5-point Likert scale ranging from 1–5 wherein 1 = very dissatisfied and 5 = very satisfied), relationship status (partnered vs. single), relationship satisfaction (5-point Likert scale), and psychosexual orientation. The latter was measured using the following question: "How would you describe your sexual orientation defined as sexual attraction, emotions, fantasies, behavior towards specified gender, or/and self-labelling in the sense of identity based on those feelings and related behaviors?". Possible answers were as follows: heterosexual, homosexual, bisexual, asexual, homoerotic, cannot define [43]. Respondents were also asked if they are engaging in RSB, defined as one or more of the following: "sexual contacts with more than one sexual partner at the same time, engaging in sexual activity with a casual person (one-night stand), frequent change of sexual partners, having intercourse with a person living with HIV, inconsistent use of condoms in oral, anal, and vaginal contacts (except for steady partner), prostitution or using the services of an escort agency, sexual contacts under influence of psychoactive substances other than alcohol and marijuana ('chemsex'), or drug injection with shared needles leading to sex, within the last 6 months" [44].

The general activation/inhibition properties of the individuals were assessed by the Behavioral Inhibition/Behavioral Activation Scale (BIS/BAS) [45]. The BAS scale is comprised of

three subscales: reward responsiveness (BAS-RR), drive (BAS-D), and fun seeking (BAS-FS). The BIS/BAS is a widely used measure of general inhibition and activation tendencies, as functional systems in the brain, with higher scores in each domain representing higher tendencies. In the present study, we used a validated Polish version of the BIS/BAS, that has been shown to have satisfactory reliability (Cronbach's $\alpha$ = 0.69–0.74) [45].

We used the 11-item Polish version of the Sexual Sensation Seeking Scale (SSSS) to assess sexual adventurism and sexual risk taking. Possible answers on the SSSS range from 1 (not at all like me) to 4 (very much like me), and the scale has been shown to have acceptable internal consistency (Cronbach's $\alpha$ = 0.83). Higher scores on the SSSS correspond with a greater tendency to sexual risk taking [46].

The 9-item Sociosexual Orientation Inventory-Revised (SOI-R) was used to measure individual restrictiveness towards a relationship. A lower level of restrictiveness is associated with a willingness to engage in uncommitted and multiple sexual relationships, whereas a higher level is associated with a predisposition towards long-term relationships [47]. The Polish version of the SOI-R has three subscales–(1) Attitude (SOI-AT), (2) Behavior (SOI-B), and (3) Desire (SOI-D). The scale has been shown to have good internal consistency with Cronbach's $\alpha$ = 0.89 [47]. Lower SOI-R scores indicate higher restrictiveness.

The Polish version of the 5-item Sexual Opinion Survey–Short Form (SOS-SF) was used to measure affective and evaluative responses to sexual stimuli, i.e., level of erotophobia/erotophilia. The Polish version of the SOS-SF has been shown to have a good reliability (Cronbach's $\alpha$ = 0.82). Higher SOS-SF scores indicate a greater erotophobic tendencies [46].

The 29-item Social Desirability Questionnaire (SDQ) was used to assess the tendency to respond in a socially desirable manner. The SDQ was based on The Marlowe-Crown Social Desirability Scale (MCSD), and validated in a Polish population with Cronbach's $\alpha$ = 0.81 [48]. Higher SDQ score reflects greater conformity to social rules and conventions.

Personality traits were measured with the Ten-Item-Personality-Inventory (TIPI). This 10-item questionnaire measures how specified personal traits apply to an individual using a 7-point scale that ranges from "strongly disagree" to "strongly agree". The following five traits are measured on the TIPI: Neuroticism, Extraversion, Openness to Experience, Agreeableness, and Conscientiousness. A Polish version of the TIPI was used, which, similar to the original scale, was shown to have an unsatisfactory Cronbach's $\alpha$ = 0.41–0.67 (the lowest for Agreeableness and Openness to Experience and the highest for Conscientiousness) [49].

The Well-Matched Relationship Questionnaire (WMRQ) was used to evaluate relationship satisfaction. The WMRQ consists of 32 questions grouped into four relationship dimensions: intimacy, disappointment, personal fulfilment, and similarity. Participants are asked to assess their relationship using a 5-point Likert scale, and higher WMRQ scores indicate a greater level of each dimension. The WMRQ has been validated and standardized in a Polish population and has been shown to have good psychometric properties (Cronbach's $\alpha$ = 0.81) [50].

## Statistical analysis

Statistical analyses were performed using Statistica 12.0 Pl computer software (StatSoft. Cracow, Poland), IBM SPSS 20 computer software with AMOS version 25 (IBM SPSS Statistics for Windows, Armonk, NY: IBM Corp; 2012), R version 4.0.2 statistics (R project accessed at: https://www.r-project.org/) with Iavaan and Semtools package (accessed at https://lavaan.ugent.be). Missing values was assessed for all variables (less than 5%). Cases with missing data were deleted when performing confirmatory factor analysis (CFA) and exploratory factor analysis (EFA). Skewness and kurtosis were assessed to check for univariate and multivariate distribution normality. Values larger than 3 for skewness or larger than 10 for kurtosis were

considered to indicate nonnormality. Mardia's coefficient, a measure of multivariate kurtosis and skewness, was used for multivariate distributions. Mardia's coefficients of $\leq 5$ were considered to be indicative of normality [37]. *P*-values of $< 0.05$ were considered to be statistically significant.

## Factor analysis

In the first step, CFA was performed using AMOS on the entire sample of women (N = 509) to analyze the fit indices in the existing model [7]. The following values of indices were used to indicate acceptable fit: the probability of a close fit (p Close Fit) = $> 0.05$; Comparative Fit Index (CFI) $\geq 0.95$, and root-mean-square error of approximation (RMSEA) $\leq 0.06$ [37]. Modification indices were inspected to identify non-fitting items. The original models did not reach satisfactory fit indices (CFI = 0.77, RMSEA = 0.065). Given the lack of multivariate normality in the data, we used EFA to investigate the structure of the model. EFA was performed using the principal axis with Promax rotation preceded by Monte-Carlo parallel analysis, as recommended by Swami et al. [37]. We also used CFA to examine model fit, using maximum likelihood method with bootstrapping. To apply EFA and CFA, we first randomly divided the study sample into two roughly equal groups using SPPS software: group 1 (N = 255) and group 2 (N = 254). EFA was performed on group 1, and CFA of the new model was performed on group 2 [37,51]. Items with factor loadings $\geq 0.40$ were retained in the model [12,37,52]. Factorability was measured using inter-item correlations ($\geq 0.50$), the Kaiser-Meyer-Olkin (KMO) measure of sampling adequacy ($\geq 0.80$), and Bartlett's test of sphericity ($p > 0.05$). An eight-factor model was developed. As in original model, Monte Carlo analysis was performed on the extracted lower-order factors [7] and the principal axis analysis was again executed. Similarly, factors with loadings $\geq 0.40$ were retained in the model. The final model consisted of eight factors and two higher order factors. CFA was performed on the final model with modification indices analysis to identify the best model fit.

## Measurement invariances

Comparisons among models were performed using AMOS plugins [53], to determine whether the SISII-W-PL can be used in different populations. Similar to an analysis performed by Velten et al. [36], we divided the study sample into subgroups according to the following variables: (1) age ($\leq 45$ years, younger; $> 45$ years, older); (2) education level (higher, university; lower, primary/secondary); (3) relationship status (partnered, singles); and (4) type of questionnaire (online; paper-pencil). Configural data and metrics were tested [10]. If the differences were significant, z-scores and *p*-values of the standardized subscale scores were reported, and effect size was estimated using Cohen's *d*. Small effects were defined as $d > 0.20$; medium, $> 0.50$; and large, $> 0.80$ [10].

## Reliability

Intraclass correlation coefficient (ICC) was used to assess reliability [37] and the Cronbach's α was used to determine internal consistency [37]. ICC values of $> 0.40$ reflect poor to fair agreement; 0.41–0.60, moderate agreement; 0.61–0.80, good agreement; and $> 0.80$, excellent agreement between two measurements [37]. Cronbach's α values of 0.5–0.75 indicate moderate reliability; 0.76–0.9, good reliability; and $> 0.90$, excellent reliability [54]. McDonald's omega was also calculated, with values $\geq 0.70$ indicative of satisfactory reliability [55].

## Construct validity

As in the original validation study [7], convergent and discriminant validity of the SESII-W-PL was assessed using correlations with other variables that measure proximal and distal constructs. Correlations were performed using Pearson's $r$, wherein $r$ values of $\geq 0.10$ indicate a weak effect size; $r \geq 0.30$, moderate; and $r \geq 0.50$, large effect size [56]. For dichotomous variables, the point-biserial correlation coefficient ($r_{pb}$) was calculated [57]. One-way ANOVA controlling for age with Bonferroni correction (if possible) was used to assess the relationship between SESII-W-PL and select socioeconomical and sexual behaviors variables. Partial eta square ($\eta^2$) values of $\geq 0.01$ were considered to be small effect sizes; $\eta^2 \geq .006$, medium effects; and $\eta^2 \geq 0.14$, large effects [58].

# Results

## General characteristics

The mean age of the studied women was $39.7 \pm 11.3$ years (range = 18–55). The majority of women were heterosexual (87.4%; n = 445), and the remaining 12.6% described themselves as homosexual (3.3%; n = 17), bisexual (8.3%; n = 4 2), asexual (0.40%; n = 2), or homoerotic (0.60%; n = 3). The majority of respondents identified as being Catholic (74.3%; n = 378), although only 30.3% (n = 154) reported regular church attendance. Most respondents (80.7%; n = 411) lived in cities and had a secondary education (44.6%; n = 227). Sixty-eight percent of women in the sample (n = 346) reported at least one pregnancy. Based on HADS scores, symptoms of depression and anxiety based were reported in 5.7% (n = 29) and 13.4% (n = 68) of respondents, respectively.

The analysis of sexual behavior revealed that 12.1% (n = 60) of women were single, 82.9% (n = 422) reported having a sexual partner, and 91.2% were sexually active in the last 4 weeks. Surprisingly, 14.9% (n = 76) of women reported recently engaging in RSB. Based on DSM-5 criteria, distress was present in 24.4% (n = 124) of women, and FSD was reported in 14.7% (n = 75) of women. In contrast, based on the FSFI, sexual problems were present in 32.2% of women, sexual distress (FSDS-R) in 39.7%, and distressing sexual concerns (FSFI and FSDS-R) in 24% of women. There were no differences in the examined variables between groups 1 and 2 with the exception of SOI-Desire, which was higher in group 1 as compared to group 2 (2.42 vs. 2.13, respectively, $p = 0.04$; see Table 1). There were also no differences between the groups that completed the survey online vs. paper-pencils with the exception of BMI, which was higher in the online group as compared to the paper-pencil group (23.6 vs. 22.3, respectively, $p = 0.001$; see Table 1).

## Factor analysis

The CFA conducted on a total sample of 509 women on the original model showed poor model fit (Table 2). As reported in the Method section, EFA was performed on group 1. Monte Carlo analysis showed an eight lower-factor solution, which accounted for 56.2% of the variance in the data. The KMO showed sampling adequacy for further analysis (KMO = 0.81, $\chi^2 = 3161.1$; Bartlett's test of sphericity, df = 630, $p < 0.001$). However, some items (items 8, 30, 32) were excluded from the model based on low loadings. In addition, items 13 and 29 presented loading overlap (similar loadings for Settings and Arousability, and for Concern about Sexual function and Arousal Contingency, respectively), and were subsequently excluded from the model. Of note, some items loaded onto a different subscale than in the original scale. In particular, items 24–26 moved from Arousability to Partner Characteristics, item 10 moved from Partner Characteristics to Arousability, item 28 moved from Sexual Power Dynamics to

**Table 1. Characteristics of the study sample.**

| Variable | Total sample | | Group 1 | | Group 2 | | Online surveys | | Paper-pencil surveys | |
|---|---|---|---|---|---|---|---|---|---|---|
| | N = 509 | | N = 255 | | N = 254 | | N = 289 | | N = 220 | |
| | M | SD | M | SD | M | SD | M | SD | Mean | SD |
| Age (years) | 39.77 | 11.27 | 39.97 | 11.29 | 39.56 | 11.27 | 39.78 | 11.33 | 39.74 | 11.20 |
| BMI (kg/m$^2$) | 23.08 | 4.37 | 22.80 | 4.09 | 23.35 | 4.63 | 23.59[a] | 4.81 | 22.27[a] | 3.45 |
| Age of first genital sex | 19 | 3.35 | 19.03 | 3.63 | 18.98 | 3.04 | 18.89 | 3.44 | 19.18 | 3.18 |
| The importance of sex for the respondent– 5-point Likert scale | 3.64 | 0.86 | 3.64 | 0.83 | 3.64 | 0.89 | 3.65 | 0.83 | 3.62 | 0.90 |
| Duration of relationship (years) | 14.21 | 10.64 | 14.49 | 10.70 | 13.95 | 10.59 | 14.89 | 10.50 | 13.16 | 10.78 |
| Relationship satisfaction | 4.12 | 1.23 | 4.11 | 1.22 | 4.13 | 1.25 | 4.08 | 1.32 | 4.17 | 1.08 |
| Number of lifetime male sexual partners | 3.85 | 4.16 | 3.82 | 3.95 | 3.87 | 4.37 | 4.00 | 4.41 | 3.61 | 3.74 |
| Number of lifetime female sexual partners | 0.30 | 1.07 | 0.22 | 0.78 | 0.38 | 1.29 | 0.34 | 1.16 | 0.25 | 0.90 |
| Sexual life satisfaction (5-point Likert scale) | 3.93 | 0.88 | 3.91 | 0.88 | 3.96 | 0.89 | 3.94 | 0.86 | 3.93 | 0.92 |
| Satisfaction from a partner as a lover | 3.97 | 1.04 | 3.97 | 1.03 | 3.96 | 1.04 | 3.96 | 1.05 | 3.97 | 1.02 |
| Number of vaginal intercourse per month | 8.02 | 7.56 | 7.97 | 7.68 | 8.07 | 7.46 | 7.96 | 7.50 | 8.11 | 7.68 |
| Number of mutual masturbation per month | 3.18 | 6.17 | 3.51 | 7.18 | 2.85 | 4.94 | 3.15 | 6.54 | 3.22 | 5.55 |
| Number of masturbation per month | 2.59 | 4.84 | 2.58 | 4.64 | 2.59 | 5.05 | 2.63 | 5.06 | 2.53 | 4.51 |
| HADS–anxiety | 6.30 | 3.75 | 6.50 | 3.87 | 6.09 | 3.62 | 6.37 | 3.86 | 6.18 | 3.58 |
| HADS–depression | 4.31 | 3.44 | 4.35 | 3.45 | 4.26 | 3.44 | 4.31 | 3.54 | 4.31 | 3.29 |
| FSFI–total score | 28.06 | 7.27 | 28.78 | 6.31 | 27.36 | 8.07 | 27.59 | 7.66 | 28.75 | 6.64 |
| FSDS-R–total score | 11.11 | 9.80 | 11.60 | 9.95 | 10.64 | 9.67 | 10.91 | 9.65 | 11.41 | 10.06 |
| BAS-D | 10.77 | 2.50 | 10.88 | 2.68 | 10.66 | 2.34 | 10.74 | 2.56 | 10.81 | 2.44 |
| BAS-FS | 11.30 | 2.37 | 11.54 | 2.41 | 11.08 | 2.31 | 11.52 | 2.28 | 10.99 | 2.47 |
| BAS-RR | 16.60 | 2.31 | 16.83 | 2.28 | 16.38 | 2.32 | 16.48 | 2.29 | 16.76 | 2.32 |
| BIS | 21.19 | 3.57 | 21.42 | 3.48 | 20.97 | 3.65 | 20.81 | 3.32 | 21.72 | 3.84 |
| SOS-SF | 18.72 | 6.09 | 18.03 | 5.47 | 19.35 | 6.57 | 18.92 | 5.77 | 18.42 | 6.54 |
| SDQ | 17.40 | 3.11 | 17.63 | 3.28 | 17.19 | 2.93 | 17.33 | 3.23 | 17.51 | 2.93 |
| SSSS | 27.63 | 5.74 | 28.03 | 5.91 | 27.26 | 5.56 | 27.89 | 5.39 | 27.27 | 6.20 |
| SOI-B | 1 | 0.78 | 0.99 | 0.76 | 1.01 | 0.80 | 1.08 | 0.77 | 0.88 | 0.79 |
| SOI-AT | 2.83 | 1.26 | 2.95 | 1.23 | 2.71 | 1.29 | 3.02 | 1.19 | 2.56 | 1.31 |
| SOI-D | 2.27 | 1.10 | 2.42[b] | 1.12 | 2.13[b] | 1.07 | 2.28 | 1.07 | 2.25 | 1.15 |
| SOI–total score | 6.09 | 2.67 | 6.36 | 2.61 | 5.85 | 2.71 | 6.38 | 2.57 | 5.69 | 2.77 |
| TITP-E | 10.41 | 2.90 | 10.53 | 2.81 | 10.30 | 2.99 | 10.39 | 2.93 | 10.44 | 2.88 |
| TIPI–A | 9.83 | 2.50 | 9.75 | 2.42 | 9.91 | 2.57 | 9.63 | 2.31 | 10.13 | 2.73 |
| TIPT–C | 10.18 | 2.87 | 10.34 | 2.60 | 10.02 | 3.11 | 9.87 | 2.80 | 10.61 | 2.93 |
| TIPI–N | 8.43 | 2.88 | 8.46 | 2.86 | 8.40 | 2.92 | 8.51 | 2.95 | 8.32 | 2.80 |
| TIPI-OTE | 9.25 | 2.31 | 9.23 | 2.10 | 9.26 | 2.50 | 9.24 | 2.16 | 9.26 | 2.53 |
| WMRQ-I | 25.23 | 9.16 | 24.85 | 9.28 | 25.58 | 9.06 | 25.30 | 9.03 | 25.12 | 9.39 |
| WMRQ-D | 24.30 | 8.15 | 24.61 | 8.06 | 24.01 | 8.25 | 24.82 | 8.47 | 23.50 | 7.59 |
| WMRQ-PF | 22.31 | 7.57 | 21.93 | 7.58 | 22.67 | 7.57 | 22.58 | 7.45 | 21.88 | 7.77 |
| WMRQ-S | 22.19 | 8.15 | 21.83 | 8.12 | 22.53 | 8.19 | 22.37 | 8 | 21.93 | 8.41 |
| WMRQ–total score | 105.4 | 26.74 | 104.0 | 27.02 | 106.8 | 26.49 | 105.4 | 26.99 | 105.4 | 26.46 |

[a]–paper-pencil vs. online version, $p < 0.05$;

[b]–group 1 vs. group 2,

$p < 0.05$; BMI–body mass index; RSB–risky sexual behaviors; HADS–Hospital Anxiety and Depression Scale; HADS-D–depression subscale, HADS-A–anxiety subscale; FSD–Female Sexual Dysfunction; FSFI–Female Sexual Function Index; FSDS-R–Female Sexual Distress Scale-Revised; DSM-5 –Diagnostic and Statistical Manual of Mental Disorders, 5th Edition; BAS-D–Behavioral Activation Scale–drive; BAS-FS–fun seeking; BAS-RR–Behavioral Activation Scale–reward responsiveness; BIS–Behavioral Inhibition Scale; SOS-SF–Sexual Opinion Survey–Short Form; SOI-R–Sociosexual Orientation Inventory Revised; SOI-B–Behavior domain; SOI-AT–attitude domain, SOI-D–Desire domain; SSSS–Sexual Sensation Seeking Scale; SDQ–Social Desirability Questionnaire; TIPI–Ten-Item-Personality-Inventory; TITP-E–Extraversion; TIPI-A–Agreeableness; TIPT-C–Conscientiousness; TIPI-N–Neuroticism; TIPI-OTE–Openness to Experience; WMRQ-I–Well-Matched Relationship Questioner–intimacy; WMRQ-D–Well-Matched Relationship Questioner–disappointment; WMRQ-PF–Well-Matched Relationship Questioner–personal fulfilment; WMRQ-S–Well-Matched Relationship Questioner–similarity; WMRQ–total–Well-Matched Relationship Questioner–total score.

Arousal Contingency, item 3 was moved from Setting to Arousability, and items 4 and 7 did not have negative loadings.

In the next step, EFA was executed on the eight-factor model to extract higher-order factors, following the original validation study [7]. Two higher-order factors–sexual excitation and sexual inhibition–were subsequently extracted. However, the Setting subscale did not reach adequate loading ($\geq 0.40$) and was subsequently removed from the model. The model had sampling adequacy for further analysis (KMO = 0.72, $\chi2$ = 635.7; Bartlett's test of sphericity, df = 28, $p < 0.001$) and explained 58.95% of the variance in the data. The CFA was executed then on this model. During the analysis, the following changes were made to reach sufficient model fit: (1) item 1 from the Relationship Importance subscale had low loading and was removed; (2) items 27 and 28 from the Sexual Power Dynamics subscale had low loading ($< 0.40$) and were thus removed. Modification indices were also checked and corrected. The final model consisted of 26 item with 7 domains and 2 higher-order factors. Please see Table 3 and Fig 2 for details, and Table 2 for fit indices. As expected, all subscales were correlated on the low or moderate level (Table 4).

**Table 2. Model fit indices for different model and different groups.**

| Model | $\chi^2$ | df | CFI | $p$ Close Fit | RMSEA (90% CI) | ΔCFA | ΔRMSEA |
|---|---|---|---|---|---|---|---|
| Single Group CFA | | | | | | | |
| Original model | 1824.23 | 581 | 0.77 | 0.0001 | 0.065 (0.061–0.075) | | |
| Polish model | 693.69 | 285 | 0.93 | 0.15 | 0.050 (0.048–0.058) | | |
| Single individuals | 673.54 | 285 | 0.90 | 0.148 | 0.053 (0.048–0.058) | | |
| Partnered individuals | 653.30 | 285 | 0.91 | 0.04 | 0.056 (0.050–0.062) | | |
| Younger ($M_{age}$ = 30.39 years; SD = 7.31) | 606.37 | 285 | 0.91 | 0.04 | .060 (0.057–0.07) | | |
| Older ($M_{age}$ = 49.79 years; SD = 3.31) | 511.63 | 285 | 0.91 | 0.03 | .059 (0.051;.067) | | |
| Lower education | 563.93 | 285 | 0.91 | 0.04 | .057 (0.050;.064) | | |
| Higher Education | 532.78 | 285 | 0.90 | 0.01 | .060 (0.053;.072) | | |
| Online version | 480.97 | 285 | 0.92 | 0.06 | .054 (0.049–0.068) | | |
| Paper-pencil version | 589.22 | 285 | 0.91 | 0.02 | .059 (0.053–0.066) | | |
| Multi Group CFA | | | | | | | |
| (1) Paper-pencil vs. online | | | | | | | |
| Configural invariance | 1290.8 | 582 | 0.924 | | 0.059 (0.053–0.067) | | |
| Weak/metric invariance | 1317.6 | 606 | 0.923 | | 0.058 (0.054–0.062) | /0.001/ | /0.001/ |
| Strong/threshold invariance | 1346.9 | 623 | 0.921 | | 0.058 (0.054–0.062) | /0.002/ | /0.000/ |
| (2) Younger vs. Older | | | | | | | |
| Configural invariance | 1330.3 | 582 | 0.911 | | 0.061 (0.057–0.066) | | |
| Weak/metric invariance | 1377.6 | 606 | 0.908 | | 0.061 (0.057–0.066) | /0.003/ | /0.000/ |
| Strong/threshold invariance | 1433.6 | 623 | 0.907 | | 0.061 (0.057–0.066) | /0.001/ | /0.000/ |
| (3) Single vs. Partnered | | | | | | | |
| Configural invariance | 1334.8 | 582 | 0.923 | | 0.061 (0.058–0.065) | | |
| Weak/metric invariance | 1368.3 | 606 | 0.921 | | 0.060 (0.055–0.064) | /0.002/ | /0.001/ |
| Strong/threshold invariance | 1398.8 | 623 | 0.918 | | 0.060 (0.055–0.064) | /0.003/ | /0.000/ |
| (4) Lower education vs. higher education | | | | | | | |
| Configural invariance | 1323.0 | 582 | 0.925 | | 0.061 (0.058–0.065 | | |
| Weak/metric invariance | 1353.4 | 606 | 0.923 | | 0.060 (0.055–0.064) | /0.002/ | /0.001/ |
| Strong/threshold invariance | 1389.4 | 623 | 0.919 | | 0.060 (0.055–0.064) | /0.004/ | /0.000/ |

df–degrees of freedom; $p$ Close Fit–the probability of a close fit; CFI–Comparative Fit Index; RMSEA–root-mean-square error of approximation.

**Table 3. Factor loadings and wording of the original version of the SESIIW and the Polish translation in group 2.**

| Load# | | Original version/ Polish translation | Load## |
|---|---|---|---|
| | | Subscales and questions | |
| 0.79 (SE) | | Arousability (SE)/ Łatwość uzyskania podniecenia (SE) | 0.84 (SE) |
| - | ENQ3 | Having sex in a different setting than usual is a real turn on for me. | 0.54 |
| | PLQ2 | Podnieca mnie seks w innych niż zazwyczaj miejscach. | |
| | Note: | This question was originally in the Setting subscale, but was moved to Arousability based on EFA loadings | |
| - | ENQ10 | Seeing a partner doing something that shows his/her talent can make me very sexually aroused. | 0.44 |
| | PLQ6 | Oglądanie partnera wykonującego czynności ukazujące jego talent wpływa na mnie bardzo podniecająco. | |
| | Note: | This question was originally in the Partner Characteristics subscale, but was moved to Arousability based on EFA loadings | |
| 0.32 | ENQ15 | Seeing an attractive partner's naked body really turns me on. | 0.69 |
| | PLQ10 | Podnieca mnie oglądanie nagiego ciała atrakcyjnego partnera seksualnego. | |
| 0.42 | ENQ17 | Just being physically close with a partner is enough to turn me on. | 0.79 |
| | PLQ12 | Samo bliskość fizyczna partnera wystarczy mi, by się podniecić. | |
| 0.51 | ENQ19 | I get very turned on when someone really wants me sexually. | 0.68 |
| | PLQ14 | Jeśli druga osoba naprawdę mnie pragnie seksualnie, bardzo się podniecam. | |
| 0.59 | ENQ20 | Fantasizing about sex can quickly get me sexually excited. | 0.53 |
| | PLQ15 | Fantazjowanie o seksie sprawia, że szybko się podniecam. | |
| 0.64 | ENQ24 | When I think about someone I find sexually attractive, I easily become sexually aroused. | - |
| | Note: | This question was moved to the Partner Characteristics subscale based on EFA loadings | |
| 0.33 | ENQ25 | With a new partner I am easily aroused. | - |
| | Note: | This question was moved to the Partner Characteristics subscale based on EFA loadings | |
| 0.44 | ENQ26 | If I see someone dressed in a sexy way, I easily become sexually aroused. | - |
| | Note: | This question was moved to the Partner Characteristics subscale based on EFA loadings | |
| 0.59 | ENQ30 | Certain hormonal changes definitely increase my sexual arousal. | - |
| | - | Z całą pewnością zmiany hormonalne zwiększają poziom mojego podniecenia seksualnego. | |
| | Note: | Did not enter the model based on low loadings in the EFA | |
| 0.55 | ENQ32 | Sometimes I am so attracted to someone, I cannot stop myself from becoming sexually aroused. | - |
| | - | Czasami jestem tak zauroczona drugą osobą, że nie mogą się powstrzymać i podniecam się seksualnie. | |
| | Note: | Did not enter the model based on low loadings in the EFA | |
| 0.60 (SE) | | Partner Characteristics (SE)/Charakterystyka Partnera (SE) | 0.77 (SE) |
| 0.51 | ENQ5 | Someone doing something that shows he/she is intelligent turns me on. | 0.49 |
| | PLQ4 | Podniecają mnie osoby, które uchodzą za inteligentne. | |
| 0.56 | ENQ8 | If I see a partner interacting well with others, I am more easily sexually aroused. | - |
| | - | Jeżeli widzę, że partner dobrze dogaduje się z innymi osobami, łatwiej się przy nim podniecam. | |
| | Note: | Did not enter the model based on low loadings in the EFA | |
| 0.66 | ENQ10 | Seeing a partner doing something that shows his/her talent can make me very sexually aroused. | - |
| | Note: | This question was moved to the Arousability subscale | |
| 0.36 | ENQ12 | Eye contact with someone I find sexually attractive really turns me on. | 0.66 |
| | PLQ8 | Podnieca mnie kontakt wzrokowy z osobą, którą uważam za atrakcyjną seksualnie. | |
| - | ENQ24 | When I think about someone I find sexually attractive, I easily become sexually aroused. | 0.51 |
| | PLQ19 | Jeśli myślę o kimś dla mnie atrakcyjnym seksualnie, łatwo się podniecam. | |
| | Note: | Based on EFA loadings, this question was moved to the Partner Characteristics subscale from the Arousability subscale | |
| - | ENQ25 | With a new partner I am easily aroused. | 0.88 |
| | PLQ20 | Łatwo się podniecam z nowym partnerem. | |
| | Note: | Based on EFA loadings, this question was moved to the Partner Characteristics subscale from the Arousability subscale | |
| - | ENQ26 | If I see someone dressed in a sexy way, I easily become sexually aroused. | 0.77 |
| | PLQ21 | Jeżeli widzę kogoś ubranego w seksowny sposób, łatwo się podniecam. | |
| | Note: | Based on EFA loadings, this question was moved to the Partner Characteristics subscale from the Arousability subscale | |

*(Continued)*

**Table 3.** (Continued)

| Load# | | Subscales and questions | Load## |
|---|---|---|---|
| | | **Original version/ Polish translation** | |
| 0.53 (SE) | | Sexual Power Dynamics (SE)/ Dynamika dominacji seksualnej (SE) | 0.72 (SE) |
| 0.54 | ENQ2 | It turns me on if my partner "talks dirty" to me during sex. | 0.59 |
| | PLQ1 | Podnieca mnie, jak partner świntuszy podczas seksu. | |
| 0.59 | ENQ6 | Feeling overpowered in a sexual situation by someone I trust increases my arousal. | 0.63 |
| | PLQ4 | Poczucie bycia zdominowaną w sytuacji seksualnej przez osobę, której ufam, podnosi poziom mojego podniecenia. | |
| 0.53 | ENQ27* | If a partner is forceful during sex, it reduces my arousal. | - |
| | - | Jeśli partner jest stanowczy podczas seksu, łatwo tracę podniecenie. | |
| | Note: | Did not enter the model based on low loadings in the EFA | |
| 0.43 | ENQ28 | Dominating my partner sexually is arousing to me. | - |
| | - | Podnieca mnie dominowanie seksualne nad partnerem. | |
| | Note: | Did not enter the model based on low loadings in the EFS | |
| 0.51 (SE) | | Smell (SE)/ Zapach (SE) | 0.71 (SE) |
| 0.68 | ENQ22 | Particular scents are very arousing to me. | 0.72 |
| | PLQ17 | Podniecają mnie określone zapachy. | |
| 0.84 | ENQ23 | Often just how someone smells can be a turn on. | 0.77 |
| | PLQ18 | Często sam zapach drugiej osoby może działać na mnie podniecająco. | |
| 0.43 (SE) | | Setting (unusual or unconcealed) (SE)/ Miejsce (nietypowe) (SE) | - |
| 0.77 | ENQ3 | Having sex in a different setting than usual is a real turn on for me. | - |
| | Note: | This question was moved to the Arousability subscale based on EFA loadings | |
| -0.32 | ENQ4* | If it is possible someone might see or hear us having sex, it is more difficult for me to get aroused. | - |
| | - | Świadomość, że ktoś może nas zobaczyć lub usłyszeć podczas seksu sprawia, że trudniej jest mi się podniecić. | |
| | Note: | Did not enter the model based on low EFA loadings in higher-order factor analysis | |
| -0.56 | ENQ7* | I find it harder to get sexually aroused if other people are nearby. | - |
| | - | Trudniej mi uzyskać podniecenie, jeśli inne osoby są w pobliżu. | |
| | Note: | Did not enter the model based on low EFA loadings in higher-order factor analysis | |
| 0.55 | ENQ13 | I get really turned on if I think I may get caught while having sex. | - |
| | - | Bardzo się podniecam, gdy myślę, że mogę zostać przyłapana podczas seksu. | |
| | Note: | Did not enter the model based on EFA, and demonstrated cross-loading with Arousability | |
| 0.38 (SI) | | Relationship Importance (SI)/Znaczenie Związku (SI) | 0.67 (SI) |
| 0.46 | ENQ1 | If I think that a partner might hurt me emotionally, I put the brakes on sexually. | - |
| | - | Jeśli spodziewam się, że partner mógłby mnie zranić emocjonalnie, wstrzymuje nasze życie erotyczne. | |
| | Note: | Did not enter the model based on low loadings in the EFA | |
| 0.54 | ENQ11 | It would be hard for me to become sexually aroused with someone who is involved with another person. | 0.66 |
| | PLQ7 | Ciężko by mi było podniecić się seksualnie, kiedy miałabym świadomość, że partner jest zaangażowany w relację z inną osobą. | |
| 0.57 | ENQ14 | If I think that I am being used sexually it completely turns me off. | 0.79 |
| | PLQ9 | Jeśli czuję, że ktoś mnie wykorzystuje seksualnie, podniecenia seksualne ustępuje natychmiast. | |
| 0.54 | ENQ16 | It is easier for me to become aroused with someone who has "relationship potential". | 0.62 |
| | PLQ11 | Łatwiej jest mi się podniecić przy osobie, która jest „materiałem na partnera". | |
| 0.54 | ENQ21 | If I am uncertain about how my partner feels about me, it is harder for me to get aroused. | 0.52 |
| | PLQ16 | Jeżeli jestem niepewny uczuć mojego partnera, trudniej jest mi się podniecić. | |
| 0.61 | ENQ33 | I really need to trust a partner to become fully aroused. | 0.54 |
| | PLQ23 | By uzyskać pełne podniecenie, muszę ufać parterowi. | |
| 0.65 (SI) | | Arousal Contingency (SI)/ Zdolność uzyskania i utrzymania podniecenia (SI) | 0.75 (SI) |
| 0.51 | ENQ34 | It is difficult for me to stay sexually aroused. | 0.58 |
| | PLQ24 | Trudno jest mi utrzymać stan podniecenia seksualnego. | |

*(Continued)*

**Table 3.** (Continued)

| Load# | | Subscales and questions | Load## |
|---|---|---|---|
| | | **Original version/ Polish translation** | |
| 0.68 | ENQ35 | When I am sexually aroused the slightest thing can turn me off. | 0.68 |
| | PLQ25 | Jeżeli jestem podniecona, nawet najdrobniejsza rzecz potrafi mnie rozproszyć tak, że tracę podniecenie. | |
| 0.71 | ENQ36 | Unless things are "just right" it is difficult for me to become sexually aroused. | 0.44 |
| | PLQ26 | Jeśli wszystko nie jest po mojej myśli, jest mi trudno się podniecić. | |
| 0.66 (SI) | | Concerns about Sexual Function (SI)/Zaniepokojenie wydolnością seksualną (SI) | 0.79 (SI) |
| 0.39 | ENQ9 | If I am concerned about being a good lover, I am less likely to become aroused. | 0.71 |
| | PLQ5 | Jeśli martwię się tym, czy jestem dobrą kochanką, trudniej mi się podniecić. | |
| 0.59 | ENQ18 | If I think about whether I will have an orgasm, it is much harder for me to become aroused. | 0.74 |
| | QPL13 | Jeśli myślę o tym, czy będę mieć orgazm, znacznie trudniej jest mi się podniecić. | |
| 0.50 | ENQ29 | Sometimes I feel so "shy" or self-conscious during sex that I cannot become fully aroused. | - |
| | - | Czasami czuje się tak bardzo „nieśmiała" czy skrępowana podczas seks, że nie mogę w pełni się podniecić. | |
| | Note: | Did not enter the model based on EFA and demonstrated cross-loading with Arousal Contingency | |
| 0.64 | ENQ31 | If I am worried about taking too long to become aroused, this can interfere with my arousal. | 0.66 |
| | PLQ22 | Jeśli martwię się tym, że mogę potrzebować zbyt dużo czasu by się podniecić, wpływa to negatywnie na moje podniecenie. | |

*–reverse scored

QEN–English original question; QPL–Polish Translation; SE–Sexual Excitation Scale; SI–Sexual Inhibition Scale; Load–Loadings

#–original version

##–Polish version.

## Multigroup comparison

The multigroup comparison (Table 2) showed structural invariance of the model across survey versions (i.e., paper-pencil vs. online; $p = 0.29$), relationship groups (single vs. partnered, $p = 0.14$), and age groups ($p = 0.06$). The model was not invariant across education groups (higher vs lower, $p = 0.03$). As compared to respondents with lower education, those with higher education had higher scores in Arousability ($p = 0.001$, $d = -0.28$), Partner Characteristics ($p = 0.001$, $d = -0.42$), Smell ($p = 0.001$, $d = -0.32$), Sexual Power Dynamics ($p = 0.04$, $d = -0.23$), and had overall higher SE scores ($p = 0.001$, $d = -0.39$). However, the effect sizes for these educational differences were in the medium range. Furthermore, the multi-group CFA comparison revealed fit indices and a drop of model fit between models that was below a $\Delta$CFI 0.010 and a $\Delta$RMSEA 0.015, that are indicative for models structural invariance [36]. Thus, the obtained result proved that the model was threshold measurement invariant across paper-pencil and online versions, single and partnered women, older and younger participants, as well as individuals with and without a university degree (Table 2).

## Reliability

Cronbach's α for each subscale and for the two higher-order factors were in the moderate to good range, with α = 0.62 for Concern about Sexual Function scale and α = 0.88 for the Smell subscale (Table 5), indicating good internal consistency. However, McDonald's omega for Concern about Sexual Function was unsatisfactory.

Eighty-four women participated in the retest procedure. The mean age of the retest subgroup was 30.3 ± 8.24 years (range = 18–44 years). All respondents in the retest subgroup were heterosexual. The mean time between the two measurements was 4.6 ± 1.7 weeks (range = 2–8

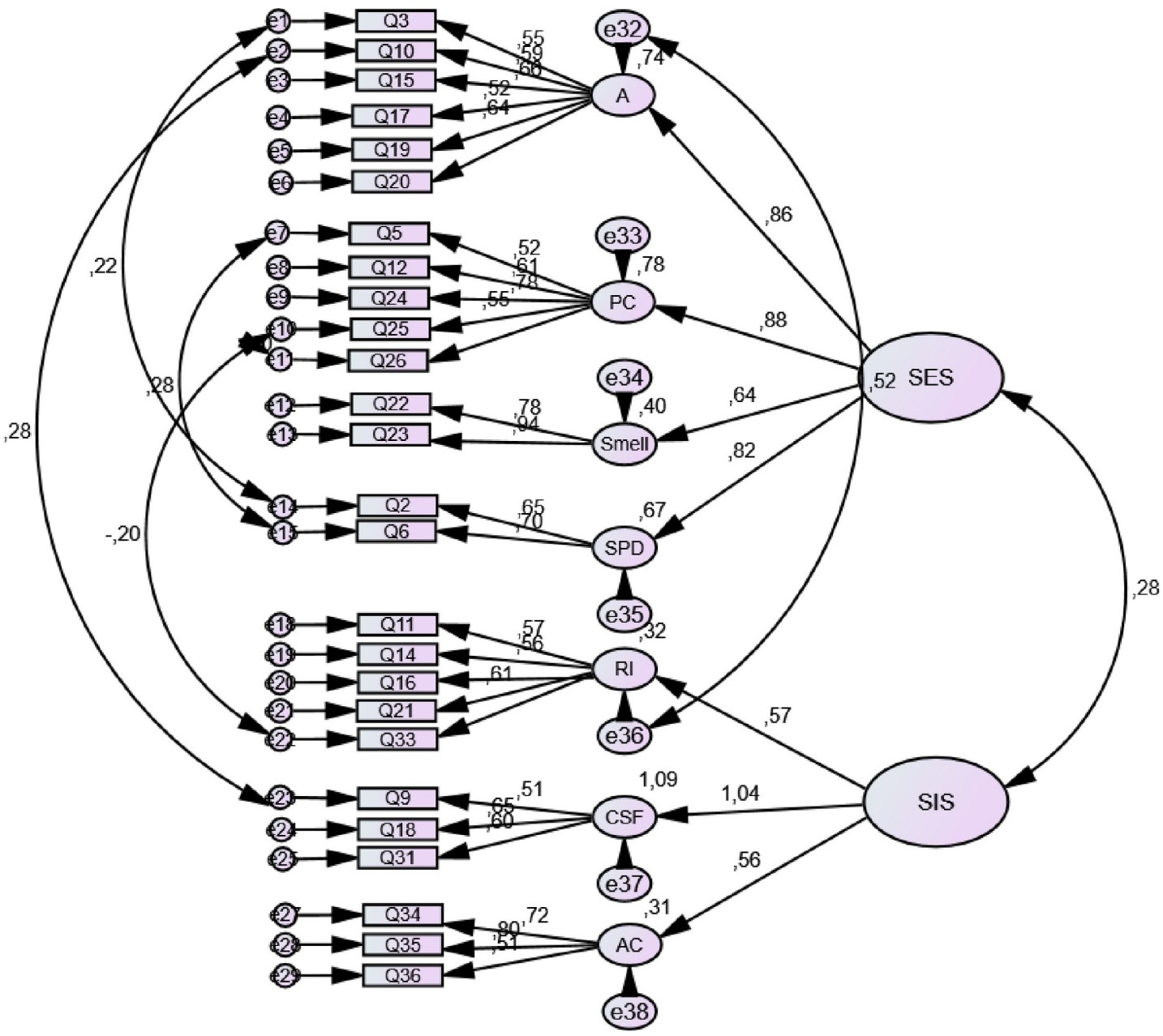

**Fig 2. Final model of the SESII-W-PL.**

weeks). The test-retest analysis (Table 5) showed good to excellent agreement between measurements, with statistically significant correlations.

## Construct validity

To measure construct validity, Pearson *r* correlations were calculated between the SESII-W-PL scales and other variables (see Table 6 in S1 Table in the Supplemental Material). As expected, total BAS scores–reflecting general activation propensity–was positively correlated with SE scores. Conversely, total scores on the BIS–reflecting inhibitory tone–were positively

**Table 4. Interrelation between SESII-W-PL subscales (Pearson's *r* correlations).**

| | Arousability | Partner Characteristics | Smell | Sexual Power Dynamics | Relationship Importance | Concerns about Sexual Function | Arousal Contingency |
|---|---|---|---|---|---|---|---|
| Arousability | 1 | | | | | | |
| Partner Characteristics | 0.523** | 1 | | | | | |
| Smell | 0.460** | 0.456** | 1 | | | | |
| Sexual Power Dynamics | 0.577** | 0.494** | 0.292** | 1 | | | |
| Relationship Importance | 0.319** | 0.192** | 0.149** | 0.154** | 1 | | |
| Concerns about Sexual Function | 0.178** | 0.252** | 0.136** | 0.122** | 0.364** | 1 | |
| Arousal Contingency | -0.052 | 0.078 | 0 | -0.035 | 0.239** | 0.415** | 1 |

\* $p < 0.05$

\*\* $p < 0.01$.

correlated with SI scores. Similarly, and as expected, SOS-SF–a marker of erotophobic tendencies–was negatively correlated with SE and positively correlated with SI. In contrast, SOI-R–a measure of individual restrictiveness to relationship–was positively correlated with SE and negatively correlated with SI. That is, women who reported being less restrictive in engaging in uncommitted relationship (i.e., lower SOI-R scores) had higher SE scores. Similarly, and as expected, SSSS scores and engaging in RSB were positively correlated with SE (but not with SI). Finally, SE was positively correlated with Extraversion and Openness to Experiences traits, and SI was negatively correlated with Neuroticism, Openness to Experiences, and Extraversion. In particular, women showing more extraverted tendencies were more likely scored higher on SI. All correlations were weak to medium.

SE subscales were not correlated with SDQ scores, indicating that responses were not altered by conformity to social rules and conventions. Thus, these results suggest that the SESII-W-PL is not influenced by social desirability. SE was not correlated with the presence of sexual distress (FSDS-R), sexual problems (FSFI), distressing sexual concerns (FSFI + FSDS-R), nor with FSD (DSM-5 criteria). However, SI was positively correlated with these distress-related variables, as well as with Concerns about Sexual Function and Arousal

**Table 5. General statistics for the lower- and higher-order factors of the SESII-W-PL.**

| Factor | TEST | | | | | RETEST | | | Cronbach's α | McDonald's omega |
|---|---|---|---|---|---|---|---|---|---|---|
| | Mean | SD | Skewness | Kurtosis | % of variance explained | Mean | SD | ICC | | |
| **Lower order** | | | | | | | | | | |
| Arousability | 3.02 | 0.65 | -0.70 | 0.22 | 19.96 | 3.08 | 0.51 | 0.7* | 0.70 | 0.71 |
| Partner Characteristics | 2.68 | 0.73 | -0.35 | -0.43 | 9.38 | 2.41 | 0.62 | 0.78** | 0.77 | 0.77 |
| Smell | 2.73 | 0.97 | -0.29 | -0.96 | 4.5 | 2.79 | 0.9 | 0.84** | 0.85 | - |
| Sexual Power Dynamics | 2.76 | 0.88 | 0.3 | -0.74 | 5.10 | 3.04 | 0.82 | 0.83** | 0.64 | - |
| Relationship Importance | 3.02 | 0.71 | -0.67 | -0.1 | 6.22 | 3.27 | 0.54 | 0.75* | 0.72 | 0.72 |
| Concerns about Sexual Function | 2.51 | 0.76 | -0.08 | -0.45 | 6.03 | 2.42 | 0.62 | 0.71* | 0.62 | 0.62 |
| Arousal Contingency | 2.36 | 0.77 | -0.09 | -0.42 | 5.02 | 2.39 | 0.8 | 0.85** | 0.70 | 0.72 |
| **Higher order** | | | | | | | | | | |
| SE | 2.80 | 0.62 | -0.42 | -0.15 | 37.33 | 2.83 | 0.45 | 0.88* | 0.76 | 0.77 |
| SI | 2.63 | 0.56 | -0.31 | -0.09 | 21.62 | 2.69 | 0.34 | 0.76* | 0.88 | 0.88 |

SE–Sexual Excitation; SI–Sexual Inhibition.

Contingency. SI was negatively correlated with Arousability, such that women with higher Arousability scores were more likely to develop sexual problems, distressing sexual concerns, and sexual dysfunction as compared to women with lower Arousability scores. Additionally, and as expected, better sexual functioning (i.e., higher FSFI scores) was positively correlated with SE and negatively correlated with SI scores. All correlations were weak.

As expected, anxiety and depression symptoms (as measured by HADS scores) were positively correlated with SI. However, the presence of depressive symptoms was not correlated with any subscale nor with the SE/SI scale. In contrast, the presence of anxiety was positively correlated with higher Arousal Contingency and SI scores.

There was a weak correlation between SE and the analyzed sexual behaviors (see Table 6 in S1 Table in the Supplemental Material), wherein higher SE scores were associated with more frequent and diverse sexual activities. Interestingly, a moderate negative correlation was noted between SE and relationship quality (assessed by WMRQ), except for in the Disappointment domain of the WMRQ.

### Associations between SESII-W-PL and demographic factors

Age was negatively correlated with all higher- and lower-order factors except Arousal Contingency (low to moderate effect). Residency, education level, and regular church attendance were not related to SE nor SI when controlling for age. Although there was a negative correlation between being currently in a relationship and Partner Characteristics, and a positive correlation between Relationship Importance and SI, these correlations did not reach statistical significance when controlling for age. When controlling for age, being sexually active in the last 4 weeks was positively correlated with Smell ($F(1, 218) = 4.39$, $p = 0.04$, $\eta^2 = 0.02$), Relationship Importance ($F(1, 218) = 5.56$, $p = 0.02$, $\eta^2 = 0.02$), and Sexual Excitation ($F(1, 218) = 4.02$, $p = 0.04$, $\eta^2 = 0.02$). These effects were medium. Similarly, being Roman Catholic was negatively correlated with Arousability ($F(12, 42) = 1.69$, $p = 0.02$, $\eta^2 = 0.23$) and positively with Concerns about Sexual Function ($F(33, 185) = 1.75$, $p = 0.01$, $\eta^2 = 0.24$). These effects were large.

### Discussion

This is the first study, to our knowledge, to evaluate the psychometric properties of the SESII-W in an Eastern and Central European sample, and to demonstrate sexual concerns in the context of the DCM. These findings have important implications for clinical practice given that, for the first time, we altered the original structure of the model to obtain a satisfying model fit. In the three prior SESII-W validation studies [10,11,13], the original structure of the model was retained, some cross-loadings between SE and SI were allowed, or unsatisfactory model fit was achieved. Based on our prior experience in cultural adaptations of questionnaires [41,42,59], we performed EFA to establish a new structure. Our approach was based on the assumption that some questions might be understood differently and contribute to other dimensions (i.e., lower order factors) of the DCM, what was described in detail in the Introduction section.

The present study included 509 women between the ages of 18 and 55 years from different socioeconomical backgrounds, applied a battery of different tests and included multigroup comparison to evaluate structural invariance. Results confirmed good psychometric validity of the new model, which included 7 lower-order and 2 higher-order factors. Given that all the examined variables were only weakly or moderately correlated with SESII-W-PL higher and lower order factors, convergent and discriminant validity of the scale was confirmed. These data indicate that the SESII-W-PL adequately measures the propensity for SE/SI. More

importantly, we demonstrated a new data on correlations between SESII-W-PL and its domain, which suggests that this scale may be useful in clinical practice.

The results of the present study showed that the Polish version of SESII-W differs from the original validation study. The Polish SESII-W has 26 items grouped into seven lower-order factors that comprise SE and SI scales. Although some items were allocated to different sub-scales as compared to the original study (for e.g., some items from the Arousability subscale were moved to the Partner Characteristics subscale and vice versa), these differences did not impact construction of higher-order factors. Of note, only the German version of the SESII-W retained both the original model structure and original number of items [10]. Indeed, the Spanish version had 34 items and a modified structure [12] and the Dutch version had 35 items and retained the original structure [11].

Ten items did not enter the Polish version of the SESII-W (i.e., items 1, 4, 7, 8, 13, 27, 28, 29, 30, and 32). In other validation studies, item 30 was either removed [12] or deemed to be problematic [10,11]. Items 28 [10] and 27 [11] were deemed to be problematic in other validation studies, but retained in the final versions of the scale. In the present validation study, items 1 ("If I think that a partner might hurt me emotionally, I put the brakes on sexually"), 13 ("I get really turned on if I think I may get caught while having sex"), and 32 ("Sometimes I am so attracted to someone, I cannot stop myself from becoming sexually aroused") were eliminated. Here, the Setting subscale was also removed, which consisted of items 4 ("Having sex in a different setting than usual is a real turn on for me") and 7 ("If it is possible someone might see or hear us having sex, it is more difficult for me to get aroused"). As reported in previous studies [31–33,59], the place and context of sexual behaviors might be more restrictive in Eastern European countries as compared to Northern European countries. In the present study, the majority of Polish women reported that they would not be willing to engage in any sexual activity in an unusual setting, or when somebody is or might even "be around". Rather, women preferred more traditional settings for sexual activities (e.g., a quiet place, bedroom rather than kitchen, locked bedroom doors, lights off). These patterns may, in part, explain the exclusion of items 13, 4, and 7 from the model. Furthermore, despite the shift from a more conservative model of relationships with a dominating male partner to equality in sexual relationship [60], many Polish women may feel uncomfortable with expressing sexuality or concerns about sexual encounters. This may be related to a lack of emotional comfort with a partner, which may be attributed to masculinity dominance in sexual education or, to be more precise, the relationship model [61]. These factors may explain, in part, the exclusion of items 1 and 32. Items 8 ("If I see a partner interacting well with others, I am more easily sexually aroused") and 29 ("Sometimes I feel so 'shy' or self-conscious during sex that I cannot become fully aroused") were also removed, which may also be due to the aforementioned cultural differences. A partner's capacity for social interaction may not be perceived as sexual cues. In contrast, performance anxiety might work as an "all or nothing" phenomenon. According to Basson's model of sexual response [62], if a woman decides to engage in sexual activities, the level of anxiety preceding the sexual act does not block her from being fully satisfied [63,64]. However, that is only one of possible explanations.

Despite the differences in the structure of the scale and the comparison of standardized scores from the four previous validation studies [7,10–12], we did not observe major differences. We also did not observe major differences between the present results and result of other studies utilizing the SESII-W in both heterosexual [18,27,29,65,66] and homosexual women [26], and between online and paper-pencil surveys. Further, Cronbach's α showed similar satisfactory values for SE, SI, Partner Characteristics, and Smell in all versions except the original validation (α for SE and SI = 0.55 and 0.70, respectively) and lesbian, gay, bisexual, transgender (LBGT) women (α for SE and SI = 0.59 and 0.60, respectively). Similar to the

present results, satisfactory α was not reached for Sexual Power Dynamics, Relationship Importance, and Concerns about Sexual Function subscales in prior studies [7,10–12,26], and Arousability was also unsatisfactory in the German validation study [10]. However, Arousal Contingency reached satisfactory reliability in all studies except the current study. Further, our test-retest analysis demonstrated good to excellent agreement between measurements with statistically significant correlations, which is consistent with prior studies [7,10–12,26].

Age was negatively correlated with all higher- and lower-order factors except Arousal Contingency. A few studies have examined the correlation between age of respondents and SE/SI. Graham et al. showed an age-related decrease in SE (but not SI) scores [7]. However, Velten et al. reported age-related decreases in both SI and SE in a German sample [15]. Interestingly, a study on a representative sample from Flanders et al. [67] used a multiple regression model to demonstrate a nonlinear association between age and SE. In that study, SE increased with age until around 25 to 40 years of age, and then steadily declined after around age 40. SI, in contrast, increased between 41 and 54 years of age and stabilized later in life [67]. A later study by the same authors extended those findings and found that SE increased with age until approximately 30 years, and subsequently decreased [68]. In contrast, SI demonstrated a U-curve shape with age, with peak SI scores reported in the 40's [68]. The SIS/SES scale was used in both studies. The relationship between SE and age might be partially explained by age-related decline in Arousability. However, inhibitory tone may also decline with age–a hypothesis that requires further investigation.

Women with sexual problems, sexual distress, and/or distressing sexual concerns demonstrated lower scores in Arousability and higher scores in SI, Arousal Contingency, and Concerns about Sexual Function. Women with sexual dysfunction also reported higher scores in SI and Concerns about Sexual Function. In contrast, women with better sexual functioning (i.e., higher FSFI scores) reported higher scores in Arousability and SE, and reported lower scores in Concerns about Sexual Function, Arousal Contingency, and SI. Similar to prior studies, the present study demonstrates that FSFI is positively correlated with SE, and negatively correlated with SI. In particular, Arousal Contingency was negatively correlated with Desire, Arousal, Lubrication, and Satisfaction subscales [16]. A study by Sanders et al. demonstrated that Arousal Contingency and Concerns about Sexual Function were positively correlated with the presence of sexual problems in general, arousal problems, orgasm problems, and with low lifetime sexual interest, evaluated by a Likert-style questionnaire [20]. A recent study by Velten et al. examined 2,275 couples in Germany using the SESII-W/M, and found that sexual functioning (assessed by the FSFI) was positively correlated with SE and negatively correlated with SI [21]. A study by Paul and Carelheira using the SESII-W/M showed that women with higher SI scores (particularly in the Inhibitory Cognition subscale) were more likely to experience a lack of sexual interest, difficulties achieving orgasm, and painful intercourses than women with lower SI scores [22]. In a small study by Hodgeson et al. [65], 15 women were asked to complete the SESII-W, and then subjective (self-reported arousal) and objective (genital region temperature) arousal was measured while participants watched an erotic film. In that study, genital temperature was positively predicted by SE and negatively predicted by SI. However, subjective arousal was negatively predicted by SE [65]. Additionally, one of largest longitudinal studies in a sample of 2,214 German women (mean age = 30.6 years) measured sexual functioning using the FSFI at baseline and after a median time of 11 months [18]. Results of that longitudinal study demonstrated that having a steady sexual partner, Arousability, Sexual Power Dynamics, and Settings were positive predictors of present and future sexual functioning, whereas depressive symptoms, Arousal Contingency, and Concerns about Sexual Function were negative predictors of current and future sexual functioning [18]. Similar results were obtained by a study of 373 women (mean age = 34.1 years) by Quinta et al. in

Spain [24,25]. Quinta et al. demonstrated that SE predicted desire, arousal, lubrication, and orgasm, as measured by the FSFI [24,25]. Only one Dutch study used DSM-IV criteria to assess FSD, and found that the presence of FSD was associated with lower total and subscale SE scores, and higher total and subscale SI scores [11]. Although we observed that the presence of FSD was only correlated with higher SI scores in the present study, this is the first study to our knowledge to apply strict DSM-5 criteria to define FSD. Given this strict criterion, we predict that level of inhibitory (but not excitatory) tone may predict FSD. However, this hypothesis must be confirmed in future studies.

Consistent with prior studies, our analysis of the construct validity of the SESII-W-PL demonstrated that higher SE/lower SI were associated with erotophilia tendencies (i.e., lower SOS-SF scores), a greater tendency for fun-seeking (i.e., higher BAS scores), and greater engagement in RSB (i.e., higher SSSS scores or self-reported RSB) [6,66,69]. Furthermore, in line with previous studies [7,10,15], women that were less restrictive in engaging in uncommitted relationship (i.e., lower SOI-R scores) had higher SE scores. However, in contrast to other studies [7,11] but in accordance with a study by Jozkowski et al. [26], we did not observe a correlation between being in relationship or relationship duration and SESII-W-PL. This null finding may be explained by the effects of age. That is, when controlling for age, the association between SESII-W-PL and relationship status reported in prior studies may be driven by age rather than by the current relationship. Interestingly, we found that higher WMRQ scores–reflecting a better relationship–were associated with lower SE. These findings are similar to results of prior studies demonstrating that women in stable relationships have lower SE scores [7,11,26,29]. These results may be interpreted as a better relationship–associated with more stability–may decrease the excitatory tone and result in lower SE. Further, given that higher similarity in a relationship, higher intimacy, and higher personal fulfilment were correlated with lower SE, we speculate that some differences between partners may be necessary for a satisfactory sexual life. Furthermore, these results challenge the hypothesis that SE and SI reflect traits tendencies, as suggested by Velten et al. [30], or that relationship quality modifies trait excitatory tone. More studies are needed to show if the observed correlation between relationship quality and SE is casual in nature.

## Clinical implications

The present study has some important clinical implications. First, we demonstrated for the first time, that the presence of sexual dysfunction assessed by strict DSM-5 criteria is correlated with greater concerns about sexual functioning and higher SI. However, the presence of sexual problems (evaluated by the FSFI screening questionnaire), sexual distress, and distressing sexual concerns (based on FSFI and FSDS-R) that do not necessarily meet DSM-5 criteria were correlated with lower Arousability and higher Arousal Contingency and SI. Based on these results, we speculate that women with a higher propensity for SI might be at risk of sexual disorders [11,20,30]. Prior research suggests that psychotherapy combined with 5-hydroxytryptamine1A receptor agonist plus Phosphodiesterase-5 inhibitors (PDE-5i) could be a potential therapeutic option [35] for women with high Arousal Continence (i.e., worries that circumstances of sexual activity will not be "perfect") and high Concerns about Sexual Function (i.e., performance anxiety). In contrast, both psychotherapy and pharmacotherapy (testosterone with PDE-5i to increase arousability) may be more appropriate for women [35] with low Arousal Contingency, low Concerns about Sexual function, and low arousability (reflecting "how easily one might become sexually aroused"). Similarly, in women with low arousability, hormonal oral contraceptives (OC's) may not be the best option as these individuals may be more susceptible to endocrine changes induced by OC's [16]. Thus, the SESII-W might be helpful in tailoring therapy for sexual problems, distress, and dysfunction.

Second, given that women with higher SE (in all domains except Smell) were more frequently engaged in RSB, those scoring high in SE should be consulted accordingly. Thus, the SESII-W might be helpful in screening for women at risk of engaging in RSB or hypersexual-related behaviors [70]. Further studies are required to confirm that hypothesis.

Third, women with lower SE reported a lower frequency and broader breadth of sexual experiences. Filling in the scale may therefore have an educational function–while reading the questions/statements women may learn about sexual responses, possible reasons for performance anxiety, and may acknowledge and accept changes in sexual responses capacity in different life stage. The SE scale, in the same mechanism, may also motivate women to seek sexually exciting situations and individual experiences to increase sexual responses in different models, not only the linear models [16], and irrespective of potential limitations (e.g. physical limitations) [27].

Finally, we found that personality type was correlated with SE and SI such that more extraverted women reported higher SE and more neurotic women reported higher SI. These findings suggest that personality type assessments may be useful for everyday clinical sexual medicine practice.

## Study limitation and future studies

The present study has some limitations. Firstly, it is not free of volunteer bias. Given that the nature of the study was somewhat intimate, some women may have felt uncomfortable and did not wish to participate. Therefore, the study sample may not fully represent all women in Poland. Selecting participant according to population structure could be helpful in future studies, but this approach cannot guarantee full representativeness. Nonetheless, the minimal required sample was achieved, suggesting that this sampling limitation does not influence the usability of the scale. Secondly, although the sample of women in the present study varied in age, there was homogeneity in relationship status. Further studies on single women are needed to better examine the model in this population. Thirdly, the study sample consisted primarily of heterosexual individuals, so the results cannot be extrapolated to homosexual populations. Fourthly, the cross-sectional nature of this study precluded us from examining the potential causality of obtained results. Finally, a recent study by Kilimnik and Meston [71] demonstrated that body esteem was negatively correlated with SI and, in women with a history of child sexual abuse, with SE. They also reported that women with low body esteem have higher SI, and a history of sexual abuse is associated with low body esteem which, in turn, decreases SE [71]. Although we did not measure history of child sexual abuse in the present study, the impact of child sexual abuse history is beyond the scope of the present study and represents a possible area for further investigation. Further, a study by Velten et al. [10] showed a negative correlation between BMI and SI domain. We did not find that correlations in the present study. Despite these limitations, the present study was demonstrated to have adequate power to perform the analyses and examined the potential associations between SESII-W-PL and the presence of FSD, based on DSM-5 criteria. Additionally, a large battery of several related measures was used to test for convergent and divergent validity of the SESII-W-PL. Finally, a multigroup comparison enabled the evaluation of the final model for structural invariance. For these reasons, we believe that the current version of the SESII-W-PL can be used in the population of Polish women between the ages of 18–55 years and provides new data that may be useful in clinical practice.

## Conclusions

The propensity for SE and SI among Polish women is comparable to rates observed in other European and North American populations. Although different from the original scale, the

SESII-W-PL demonstrated good psychometric properties and can therefore be used in the population of Polish women between the ages of 18–55 years. Using the SESII-W-PL, we found that a higher propensity for SI correlated with the presence of sexual problems, distress, and sexual dysfunction, whereas a higher propensity for SE correlated with personality types (i.e., extraversion and openness to new experiences) and the tendency to engage in RSB. Although both inhibitory and excitatory tone was shown to decrease with age, excitatory tone may be altered by relationship quality.

## Supporting information

**S1 Table. Correlations between the SESII-W-PL and other variables (Pearson's *r* correlations).**
(DOCX)

**S1 Appendix. The Polish version of the SESII-W (SESII-W-PL).**
(DOCX)

## Author Contributions

**Conceptualization:** Krzysztof Nowosielski, Jacek Kurpisz, Robert Kowalczyk.

**Data curation:** Krzysztof Nowosielski, Jacek Kurpisz, Robert Kowalczyk.

**Formal analysis:** Krzysztof Nowosielski, Jacek Kurpisz, Robert Kowalczyk.

**Funding acquisition:** Krzysztof Nowosielski, Jacek Kurpisz, Robert Kowalczyk.

**Investigation:** Krzysztof Nowosielski, Jacek Kurpisz, Robert Kowalczyk.

**Methodology:** Krzysztof Nowosielski, Jacek Kurpisz, Robert Kowalczyk.

**Project administration:** Krzysztof Nowosielski, Jacek Kurpisz, Robert Kowalczyk.

**Resources:** Krzysztof Nowosielski, Jacek Kurpisz, Robert Kowalczyk.

**Software:** Krzysztof Nowosielski, Jacek Kurpisz, Robert Kowalczyk.

**Supervision:** Krzysztof Nowosielski, Jacek Kurpisz, Robert Kowalczyk.

**Validation:** Krzysztof Nowosielski, Jacek Kurpisz, Robert Kowalczyk.

**Visualization:** Krzysztof Nowosielski, Jacek Kurpisz, Robert Kowalczyk.

**Writing – original draft:** Krzysztof Nowosielski, Jacek Kurpisz, Robert Kowalczyk.

**Writing – review & editing:** Krzysztof Nowosielski, Jacek Kurpisz, Robert Kowalczyk.

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
