## [Decision Letter · Decision Letter 0]

16 Dec 2020

PONE-D-20-32711

Sexual Inhibition and Sexual Excitation in the Population of Polish women

PLOS ONE

Dear Dr. Nowosielski,

Thank you for submitting your manuscript to PLOS ONE. After careful consideration, we feel that it has merit but does not meet PLOS ONE’s publication criteria as it currently stands. Therefore, we invite you to submit a revised version of the manuscript that addresses the points raised during the review process.

We look forward to receiving your revised manuscript.

Kind regards,

Julia Velten

Academic Editor

PLOS ONE

Additional Editor Comments (if provided):

Both reviewers suggested some additional copy editing to help with the English language. There are a great number of grammatical errors and typos throughout the manuscript that should be corrected before resubmission.

Please consult the PLOS ONE homepage for open access requirements. All data used in this study (including data set and the new questionnaire) should be uploaded to a public repository or should be uploaded as part of the submission.

Abstract

• Please provide a p value for Chi2-test.

• Please introduce all abbreviations before first use (in the abstract and elsewhere)

• Please refrain from implying causality. The cross-sectional data does not allow for statements about causality. (e.g., one cannot determine whether relationship quality affects SE or vice versa)

• Although older participants showed lower SI/SE, one cannot deduct that these traits actually decrease. Generational differences may be an alternative explanation for this effect.

Introduction

• Line 77: Please differentiate between “psychometric properties” and “internal consistency”. Those are not the same.

• I would strongly suggest to condense the studies presented and not simply list different items and values.

• Line 164: This is not a hypothesis but rather a broad statement of different expected associations. Please rephrase.

• Also, please be more precise. Stating that “a correlation” is expected is a relatively weak hypothesis.

• Line 182: This is also not a testable hypothesis (“to observe not difference”). On what literature is this based? How do the authors testing the absence of this relationship?

Method

• Please elaborate why you chose to assess sexual dysfunctions by asking about the times the participants experienced positive sexual responses.

• The cutoffs for the FSFI and FSDS-R are not the ones usually used in the literature. Please elaborate.

• Which R packages were used for the analysis?

Results

• Please consider revising some of the very large tables in order to make the content more readable. Alternatively, you could move some of the tables (e.g., Table 6) to the supplementary materials.

Discussion

• Line 588: Why would these pharmacological agents work best under these circumstances?

• Line 596: Whether the specificity and sensitivity of the SESII-W is high enough to use it as a screening tool for risk behaviors is questionable.

• Line 645: It seems an overinterpretation of data that Polish women would be “fully satisfied” despite experiencing anxiety beforehand. Alternatively, they might have lowered their expectations, which might cause them not to expect personal fulfillment.

• Line 650: The label LGBT does not fit (G stands for gay)

Journal Requirements:

2) Please note that according to our submission guidelines (http://journals.plos.org/plosone/s/submission-guidelines), outmoded terms and potentially stigmatizing labels should be changed to more current, acceptable terminology. For example: “Caucasian” should be changed to “white” or “of [Western] European descent” (as appropriate).

3) Thank you for stating the following financial disclosure:

 [The funders had no role in study design, data collection and analysis, decision to

publish, or preparation of the manuscript.].

4) We note that you have indicated that data from this study are available upon request. PLOS only allows data to be available upon request if there are legal or ethical restrictions on sharing data publicly. For information on unacceptable data access restrictions, please see http://journals.plos.org/plosone/s/data-availability#loc-unacceptable-data-access-restrictions.

Reviewers' comments:

Reviewer's Responses to Questions

**Comments to the Author**

1. Is the manuscript technically sound, and do the data support the conclusions?

Reviewer #1: Yes

Reviewer #2: Yes

2. Has the statistical analysis been performed appropriately and rigorously? 

Reviewer #1: Yes

Reviewer #2: Yes

3. Have the authors made all data underlying the findings in their manuscript fully available?

Reviewer #1: No

Reviewer #2: Yes

4. Is the manuscript presented in an intelligible fashion and written in standard English?

Reviewer #1: No

Reviewer #2: No

5. Review Comments to the Author

Reviewer #1: Thank you to the authors for submitting this important work in language/cultural validation of the SESII-W in a sample of Polish women. This translation will be useful for future sex research in Polish-speaking samples, and allow non-Polish researchers better understand the Polish sociocultural context of sexual response and sex research. Overall, the methods and analysis are well done, though the manuscript requires significant revision to be clear and readable.

Major revisions:

Title:

- Should replace "the Population of Polish Women" with "a Sample of Polish Women"

Abstract:

- The first half of the abstract should make it more clear that the study was a validation of a Polish translation of the SESII-W in a sample of Polish women.

Introduction:

- The study would benefit from a more detailed review/summary of the dual control model and how it relates to other important clinical and research variables in sexual response, functioning, and behaviour. This should be done before talking about the specific previous translations. This will be helpful when authors comment on about how the SESII-W correlates with other sexuality measures in the hypotheses and discussion sections.

- For the descriptions of previous German, Dutch, and Spanish language validations, the authors can comment more on what implications these findings had for the current translation into Polish (for example, did the authors try to account for some of the limitations in those studies in the current study). Or, the authors might state more clearly what important patterns or implications these previous studies might have for readers in understanding the current research.

- The paragraph starting on line 124 talks about a wide range of different past findings involving the SESII-W. The authors can make it more clear what the central takeaway from these past findings are, or explain more clearly how these findings relate to the current work.

- For the description starting on line 159 of the study hypotheses - I recognize that each of these hypotheses listed do fit with what has been previously reported in the literature about the dual control model. However, in the context of the current paper they don't seem justified because the earlier parts of the paper do not focus on many of these issues. For example, I don't think the paper talks about general activation/inhibition at all until this point. A more thorough literature review (as per my previous point) would be useful. Especially useful would be more elaboration on sociocultural differences relevant to Polish women for readers who are not as familiar with the Polish sociocultural context of sexuality.

Methods:

- For the participants who returned incomplete questionnaires, were these participants excluded listwise? Was other approaches to handling this missing data considered, such as multiple imputation? Especially due to the large number of incomplete responses in the online subsample.

- For figure 1, please double check your numbers because I don't think they add up correctly.

- The Materials section introduces measurement of weight and height/BMI but this seems to be the first time this is introduced. Explanation for why BMI is being measured should be made more clear in the study hypotheses.

- Line 259, the sentence starting "Using the set of that question..." I cannot follow what is being said in this sentence.

- For the description of questionnaires, each scale should be described to the same level of detail. Some questionnaires are described in more detail than others. This should be made more consistent. For example, the authors can make sure to include for each scale what the scale assesses, how participants respond, what higher/lower scores mean, total number of items, scale range, subscales, and Cronbach's alphas.

- Are Cronbach's alphas for the questionnaires in the current sample available?

Results:

- For sexual orientations in line 391, was "asexual" assessed differently than other sexual orientations, as its the only category prefaced by "declared themselves as"? As well, the preferred English term is "same-sex oriented" or "same-sex attracted" rather than "homosexual", but in this case it would also depend on how these terms are understood in Polish; I'll differ to the editor on how sexual orientations should be described for PLOS ONE.

- In the paragraph about factor analysis starting on line 442, these results could be split into multiple paragraphs (for example, EFA can be in one paragraph and CFA in another) to be more readable.

- For the paragraph starting on line 557, it's not clear how a general pattern of weak/moderate correlation between SESII-W factors and other study variables show both convergent and discriminant validity of the instrument. Also, see my point below.

- In general, the results section should focus just on the specific outcomes of the analyses. Interpretations, such as about the validity of the instrument, the novelty of certain findings, or the potential impact of one variable on another at a conceptual level, should be reserved for the discussion section.

Discussion:

- For the paragraphs starting on line 579, line 596, line 600, and line 606, it would be valuable to focus more on how the patterns observed in the current study impact our understanding of the validity and reliability of your translated instrument. It is less useful to focus on how sexual excitation and inhibition more generally predict specific interventions or treatment options for sexual difficulties here as it detracts from the main message of the results, may be too speculative, and feels beyond the scope of the current study. Potential utility of the SESII-W in future studies or clinical work may warrant its own paragraph later on in the discussion section.

- Line 580 - what does "censers" mean?

- For the paragraph starting on line 623, it's not clear how some of the sociocultural differences proposed by the authors might directly relate to response patterns on specific questions. For example, how might lower average levels of sexual liberalism in Central Europe result in items 13, 4, and 7 not being valid anymore. The authors can provide a more detailed description of their proposed mechanism or rationale for these differences.

- The discussion would benefit from more explanation of how the need to drop or move certain questions or other ways in which the questionnaire needed to be adjusted to ensure good model fit may or may not impact its usability or interpretability of the Polish instrument. For example, should future researchers be cautious in applying the current instrument to certain contexts given these issues in replicating the original English-language model fit? Or, perhaps these results have implications for the nature of the SESII-W scale or the dual control model more generally?

- For the discussion of age starting on line 660, do the authors think this have any significance to the usability of the Polish instrument?

- For the discussion of sexual problems starting on line 673, it would be useful to describe in more detail the implications the current findings have with regards to the validity of the Polish instrument. As it is currently, it reads more like a summary of the literature on the association between the SESII-W and sexual functioning, which would be a better fit for the introduction. Also, this paragraph I think is the only one to report specific r values when reviewing previous literature, this should be made consistent with the rest of the manuscript.

- For the discussion on study limitations, it would be useful to describe potential ways of addressing some of the limitations raised (e.g., representativeness of the sample) in future studies, as well as any restrictions these limitations may place on which populations the current Polish instrument is valid in. Furthermore, it is not clear how issues such as the role of child sexual abuse applies to the current study (i.e., are they really limitations or just areas for future research that are outside the scope of the current study?).

Editing:

- The manuscript can benefit from further editing for fluency/clarity of English writing. Three common issues I noticed throughout the manuscript were:

- The SESII-W is often mis-spelled as SISII-W

- The acronym for risky sexual behaviours (RSB) is sometimes misspelled as RBS

- Geographical regions should be rendered consistently: for example, "Central Europe" or "Eastern Europe", and other regions such as "Northern Europe", "Western Europe", and "North America", should all be written/spelled consistently throughout the manuscript.

- There are numerous other small spelling errors throughout the manuscript.

Reviewer #2: The present study presents a validation study of the SESII-W in a population of women living in Poland. The topic of the study is relevant and significantly contributes to the current state of research. However, before publication can be recommended some issues should be clarified.

First of all there are many typological and grammatical errors in the manuscript. The manuscript should definitely be cross-checked by a native English speaking person.

The whole manuscript is quite long and is missing a clear structure a some points. The manuscript should be at least three to four pages shorter. For example lines 63 to 123 could be much shorter and the whole discussion is quite long. The authors could also think about reporting about less questionnaires in the external validation because it does not become clear why so many questionnaires are needed, many of them measuring closely related constructs. Furtermore, I would delete the whole hypothesis section and rather add a two-sentence paragraph summarizing the main aims of the study.

Concerning the study procedures I was asking myself why informed consent was only achieved verbally in the paper-pencil-version participants. Furthermore, what were the precise contents of the consent form. Where did the participants from the paper-pencil-group fill out the questionnaires? At home or at the doctor's office? How much time did the study take for each participant? Were the participants that dropped out from the online version different from those with the paper-pencil version since the drop-out numbers were considerably higher in the online version. It does not become clear whether or not the semi-structured interview questions concerning the evaluation of FSD DSM-5 criteria were included only in the paper-pencil version. It should become clear whether or not the online and paper-pencil-versions were exactly comparable.

At some points of the statistical methods section the authors provide some information that belong in the results, for example lines 342 to 343.

Some of the tables in the results appear quite huge and thus confusing, especially table 6. This is clearly also a consequence of the large amount of questionnaires applied within the present study.

The discussion is at some points way too long and lacks a clear structure. For example, the second to fifth paragraph in the discussion should rather be at the end of the manuscript as they provide some conclusions based on the study findings. I would suggest to structure the discussion as follows: 1.) Short summary of the main study aim with one sentence about the innovativeness of the present study, 2.) half a page summary of the main study findings (without presenting precise numbers), 3.) more in-depth discussion of the study findings in comparison to previous research, 4.) Study limitation, 5.) Future studies, 6.) Main conclusions.

6. PLOS authors have the option to publish the peer review history of their article (what does this mean?). If published, this will include your full peer review and any attached files.

Reviewer #1: No

Reviewer #2: No

---

## [Author Response · Author response to Decision Letter 0]

12 Jan 2021

Dear reviewer,

Thank you for your valuable comments. The answer to all remarks is listed below.

Additional Editor Comments (if provided):

1. Both reviewers suggested some additional copy editing to help with the English language. There are a great number of grammatical errors and typos throughout the manuscript that should be corrected before resubmission.

Done

2. Please consult the PLOS ONE homepage for open access requirements. All data used in this study (including data set and the new questionnaire) should be uploaded to a public repository or should be uploaded as part of the submission.

Done

3Abstract

• Please provide a p value for Chi2-test.

Done

4. Please introduce all abbreviations before first use (in the abstract and elsewhere)

Done

5. Please refrain from implying causality. The cross-sectional data does not allow for statements about causality. (e.g., one cannot determine whether relationship quality affects SE or vice versa)

Corrected

6. Although older participants showed lower SI/SE, one cannot deduct that these traits actually decrease. Generational differences may be an alternative explanation for this effect.

Corrected

7. Introduction

• Line 77: Please differentiate between “psychometric properties” and “internal consistency”. Those are not the same.

Corrected 

8. I would strongly suggest to condense the studies presented and not simply list different items and values.

Corrected 

9. Line 164: This is not a hypothesis but rather a broad statement of different expected associations. Please rephrase.

Corrected

10. Also, please be more precise. Stating that “a correlation” is expected is a relatively weak hypothesis.

Corrected 

11. Line 182: This is also not a testable hypothesis (“to observe not difference”). On what literature is this based? How do the authors testing the absence of this relationship?

Corrected. The measurement invariance was tested, similar to the latest study by Velten et al (please see position 30 in the references).

12. Method

• Please elaborate why you chose to assess sexual dysfunctions by asking about the times the participants experienced positive sexual responses.

The scale will be also used in assessing sexual function in another study. That was the reason to as about sexual function and dysfunction in a positive way. 

13. The cutoffs for the FSFI and FSDS-R are not the ones usually used in the literature. Please elaborate.

Corrected. That cut-offs are based on the validation on Polish versions of both scales – please see ref 35 and 36. 

14. Which R packages were used for the analysis?

R version 4.0.2 was used. That information was added. 

15. Results

• Please consider revising some of the very large tables in order to make the content more readable. Alternatively, you could move some of the tables (e.g., Table 6) to the supplementary materials.

The table was moved to supplementary materials. 

16. Discussion

• Line 588: Why would these pharmacological agents work best under these circumstances?

Please see Bloemers J, van Rooij K, Poels S, Goldstein I, Everaerd W, Koppeschaar H, et al. Toward personalized sexual medicine (part 1): integrating the "dual control model" into differential drug treatments for hypoactive sexual desire disorder and female sexual arousal disorder. J Sex Med. 2013;10(3):791-809. Epub 2012/11/08. doi: 10.1111/j.1743-6109.2012.02984.x. PubMed PMID: 23130782. – from more details. As the current study does not concern pharmacotherapy, that subject was not deliberated on. 

•17. Line 596: Whether the specificity and sensitivity of the SESII-W is high enough to use it as a screening tool for risk behaviors is questionable.

Thank you for that remark. We agree that this study does not support that statement. For that reason, an additional sentence “Further studies are required to confirm that hypothesis” was added. 

18. Line 645: It seems an overinterpretation of data that Polish women would be “fully satisfied” despite experiencing anxiety beforehand. Alternatively, they might have lowered their expectations, which might cause them not to expect personal fulfillment.

Thank you for your comments. To meet your suggestion the sentence was modified – “if a woman decide to engage in sexual activities, according to Basson model of sexual response[58], the level of anxiety preceding sexual act does not block her from being fully satisfied[59, 60]. However, that is only one of possible explanations”. 

19. Line 650: The label LGBT does not fit (G stands for gay)

Thank you for that remark. LGBT was change to lesbian once

Journal Requirements:

Checked 

2) Please note that according to our submission guidelines (http://journals.plos.org/plosone/s/submission-guidelines), outmoded terms and potentially stigmatizing labels should be changed to more current, acceptable terminology. For example: “Caucasian” should be changed to “white” or “of [Western] European descent” (as appropriate).

Corrected

3) Thank you for stating the following financial disclosure:

 [The funders had no role in study design, data collection and analysis, decision to

publish, or preparation of the manuscript.].

a. Please clarify the sources of funding (financial or material support) for your study. List the grants or organizations that supported your study, including funding received from your institution.

d. If you did not receive any funding for this study, please state: “The authors received no specific funding for this work.”

4) We note that you have indicated that data from this study are available upon request. PLOS only allows data to be available upon request if there are legal or ethical restrictions on sharing data publicly. For information on unacceptable data access restrictions, please see http://journals.plos.org/plosone/s/data-availability#loc-unacceptable-data-access-restrictions.

Reviewers' comments:

Reviewer's Responses to Questions

Comments to the Author

Dear reviewers, 

Thank you for your time spent reviewing the MS and all your valuable comments. The answers are listed below. 

Reviewer #1: Thank you to the authors for submitting this important work in language/cultural validation of the SESII-W in a sample of Polish women. This translation will be useful for future sex research in Polish-speaking samples, and allow non-Polish researchers better understand the Polish sociocultural context of sexual response and sex research. Overall, the methods and analysis are well done, though the manuscript requires significant revision to be clear and readable.

Major revisions:

Title:

- Should replace "the Population of Polish Women" with "a Sample of Polish Women"

Changed as requested

Abstract:

- The first half of the abstract should make it more clear that the study was a validation of a Polish translation of the SESII-W in a sample of Polish women.

Changed as suggested

Introduction:

- The study would benefit from a more detailed review/summary of the dual control model and how it relates to other important clinical and research variables in sexual response, functioning, and behaviour. This should be done before talking about the specific previous translations. This will be helpful when authors comment on about how the SESII-W correlates with other sexuality measures in the hypotheses and discussion sections.

Changed as suggested. 

- For the descriptions of previous German, Dutch, and Spanish language validations, the authors can comment more on what implications these findings had for the current translation into Polish (for example, did the authors try to account for some of the limitations in those studies in the current study). Or, the authors might state more clearly what important patterns or implications these previous studies might have for readers in understanding the current research.

Corrected as suggested 

- The paragraph starting on line 124 talks about a wide range of different past findings involving the SESII-W. The authors can make it more clear what the central takeaway from these past findings are, or explain more clearly how these findings relate to the current work.

Corrected as suggested 

- For the description starting on line 159 of the study hypotheses - I recognize that each of these hypotheses listed do fit with what has been previously reported in the literature about the dual control model. However, in the context of the current paper they don't seem justified because the earlier parts of the paper do not focus on many of these issues. For example, I don't think the paper talks about general activation/inhibition at all until this point. A more thorough literature review (as per my previous point) would be useful. Especially useful would be more elaboration on sociocultural differences relevant to Polish women for readers who are not as familiar with the Polish sociocultural context of sexuality.

Corrected. The sociocultural differences was discussed in the discussion section of the MS

Methods:

- For the participants who returned incomplete questionnaires, were these participants excluded listwise? Was other approaches to handling this missing data considered, such as multiple imputation? Especially due to the large number of incomplete responses in the online subsample.

The majority of incomplete questionnaire had only up to 5 first questions answered. It seemed like that participants just agreed to participated and then resigned after few questions. For that reason, neither of that totally incomplete questionaries were further analyzed. 

- For figure 1, please double check your numbers because I don't think they add up correctly.

Thank you for that remark. Corrected as suggested. 

- The Materials section introduces measurement of weight and height/BMI but this seems to be the first time this is introduced. Explanation for why BMI is being measured should be made more clear in the study hypotheses.

BMI is a standard question in all studies concerning sexual function as is some research body image and BMI were related to sexual function. However, mostly because the MS is quite long, we decided not to describe all possible association as it would take ages to follow. Furthermore, BMI was associated neither with lower no higher order factor of SESII-W

- Line 259, the sentence starting "Using the set of that question..." I cannot follow what is being said in this sentence.

That sentence was changed to “Based on that set of questions diagnosis of FSD according to DSM-5 criteria was accomplished” to be more understandable. 

- For the description of questionnaires, each scale should be described to the same level of detail. Some questionnaires are described in more detail than others. This should be made more consistent. For example, the authors can make sure to include for each scale what the scale assesses, how participants respond, what higher/lower scores mean, total number of items, scale range, subscales, and Cronbach's alphas.

Thank you for that suggestion. However, once again the detailed description of all used measurement is much beyond the scope of this study. Although to meet your exception a more detailed description of SSSS was introduced. 

- Are Cronbach's alphas for the questionnaires in the current sample available?

Yes, all are available and within the acceptable ranges. That values wasn’t, however, reported, as that would once again, increase the length of the MS. 

Results:

- For sexual orientations in line 391, was "asexual" assessed differently than other sexual orientations, as its the only category prefaced by "declared themselves as"? As well, the preferred English term is "same-sex oriented" or "same-sex attracted" rather than "homosexual", but in this case it would also depend on how these terms are understood in Polish; I'll differ to the editor on how sexual orientations should be described for PLOS ONE.

Sexual orientation was defined as sexual attraction, emotions, fantasies, behavior, self-labeling, or a combination of these. As Asexual is not a sexual orientation per se, we decided to refer that as “declared themself”. After your remark, we decided to change that to “describe as”. According to PLOS policy homosexual might be used as it does not sound offensive. 

- In the paragraph about factor analysis starting on line 442, these results could be split into multiple paragraphs (for example, EFA can be in one paragraph and CFA in another) to be more readable.

Corrected.

- For the paragraph starting on line 557, it's not clear how a general pattern of weak/moderate correlation between SESII-W factors and other study variables show both convergent and discriminant validity of the instrument. Also, see my point below.

Corrected and move to discussion section as suggested.

- In general, the results section should focus just on the specific outcomes of the analyses. Interpretations, such as about the validity of the instrument, the novelty of certain findings, or the potential impact of one variable on another at a conceptual level, should be reserved for the discussion section.

Corrected and move to discussion section as suggested.

Discussion:

- For the paragraphs starting on line 579, line 596, line 600, and line 606, it would be valuable to focus more on how the patterns observed in the current study impact our understanding of the validity and reliability of your translated instrument. It is less useful to focus on how sexual excitation and inhibition more generally predict specific interventions or treatment options for sexual difficulties here as it detracts from the main message of the results, may be too speculative, and feels beyond the scope of the current study. Potential utility of the SESII-W in future studies or clinical work may warrant its own paragraph later on in the discussion section.

Corrected as requested. 

- Line 580 - what does "censers" mean?

It was meant to be “concerns” – corrected 

- For the paragraph starting on line 623, it's not clear how some of the sociocultural differences proposed by the authors might directly relate to response patterns on specific questions. For example, how might lower average levels of sexual liberalism in Central Europe result in items 13, 4, and 7 not being valid anymore. The authors can provide a more detailed description of their proposed mechanism or rationale for these differences.

That is because based on conservative approach - women would not be willing to engage in any sexual activities if somebody is around or in any unusual settings – women in Poland prefer more “usual” settings and more “peaceful” environment (restring from sexual activities if anyone is or might be around. That was explained in the Discussion section

- The discussion would benefit from more explanation of how the need to drop or move certain questions or other ways in which the questionnaire needed to be adjusted to ensure good model fit may or may not impact its usability or interpretability of the Polish instrument. For example, should future researchers be cautious in applying the current instrument to certain contexts given these issues in replicating the original English-language model fit? Or, perhaps these results have implications for the nature of the SESII-W scale or the dual control model more generally?

Thank you for that suggestion. We added the paragraph explaining that at the beginning of discussion. 

- For the discussion of age starting on line 660, do the authors think this have any significance to the usability of the Polish instrument?

We think that it does not have any influence on the questionnaire usability mostly because the age range of our sample was quite wide (18-55). However, further studies are needed to picture the relation between age and excitation/inhibition. But that is beyond the scope of this article. 

- For the discussion of sexual problems starting on line 673, it would be useful to describe in more detail the implications the current findings have with regards to the validity of the Polish instrument. As it is currently, it reads more like a summary of the literature on the association between the SESII-W and sexual functioning, which would be a better fit for the introduction. Also, this paragraph I think is the only one to report specific r values when reviewing previous literature, this should be made consistent with the rest of the manuscript.

Modify as requested 

- For the discussion on study limitations, it would be useful to describe potential ways of addressing some of the limitations raised (e.g., representativeness of the sample) in future studies, as well as any restrictions these limitations may place on which populations the current Polish instrument is valid in. Furthermore, it is not clear how issues such as the role of child sexual abuse applies to the current study (i.e., are they really limitations or just areas for future research that are outside the scope of the current study?).

Addressed as suggested 

Editing:

- The manuscript can benefit from further editing for fluency/clarity of English writing. Three common issues I noticed throughout the manuscript were:

- The SESII-W is often mis-spelled as SISII-W

Corrected

- The acronym for risky sexual behaviours (RSB) is sometimes misspelled as RBS

Corrected

- Geographical regions should be rendered consistently: for example, "Central Europe" or "Eastern Europe", and other regions such as "Northern Europe", "Western Europe", and "North America", should all be written/spelled consistently throughout the manuscript.

Corrected 

- There are numerous other small spelling errors throughout the manuscript.

Corrected by Native English Editor.

Reviewer #2: The present study presents a validation study of the SESII-W in a population of women living in Poland. The topic of the study is relevant and significantly contributes to the current state of research. However, before publication can be recommended some issues should be clarified.

Thank you for your valuable comments. All are addressed below.

1. First of all there are many typological and grammatical errors in the manuscript. The manuscript should definitely be cross-checked by a native English speaking person.

The whole manuscript is quite long and is missing a clear structure a some points. The manuscript should be at least three to four pages shorter. For example lines 63 to 123 could be much shorter and the whole discussion is quite long.

The MS was edited by Native Editorial Team. Introduction and Discussion was modified as requested, except the description of SESII-W.

2. The authors could also think about reporting about less questionnaires in the external validation because it does not become clear why so many questionnaires are needed, many of them measuring closely related constructs. 

Thank you for that suggestion. However, to meet the validation procedure requirement, most of the scales had to be retained. 

3. Furtermore, I would delete the whole hypothesis section and rather add a two-sentence paragraph summarizing the main aims of the study.

Thank you for that suggestion. That section was modified but, as other reviewers requested, hypothesis section had to be retained. 

4. Concerning the study procedures I was asking myself why informed consent was only achieved verbally in the paper-pencil-version participants. 

The informed consent was required for both versions. If in online version one did not agree to participate, he/she for transferred to end page of the questionnaire. Please see Methods section line 187 for details. 

5. Furthermore, what were the precise contents of the consent form. 

It was in accordance to RODO requirement, standard from for all studies in Europe. 

6. Where did the participants from the paper-pencil-group fill out the questionnaires? At home or at the doctor's office? 

The questionnaire was filed at home and then returned (personally or by mail) to the office. That information was added. 

7. How much time did the study take for each participant? 

About 40 minutes. That information was added. 

8. Were the participants that dropped out from the online version different from those with the paper-pencil version since the drop-out numbers were considerably higher in the online version. 

It is impossible to say, as all that returned incomplete questionaries had fill in only a few questions on page one. That was also cleared out in the text. 

9. It does not become clear whether or not the semi-structured interview questions concerning the evaluation of FSD DSM-5 criteria were included only in the paper-pencil version. It should become clear whether or not the online and paper-pencil-versions were exactly comparable.

The version were exactly the same. To make it clear the sentence “Based on that set of questions diagnosis of FSD according to DSM-5 criteria was accomplished in both paper-pencil and online version of the scale” was added. 

10. At some points of the statistical methods section the authors provide some information that belong in the results, for example lines 342 to 343.

We cannot agree as it reflects the methods of the study – it justifies why EFA was performed. 

11. Some of the tables in the results appear quite huge and thus confusing, especially table 6. This is clearly also a consequence of the large amount of questionnaires applied within the present study.

The table was move to the supplementary materials, as suggested by other reviewer. 

12. The discussion is at some points way too long and lacks a clear structure. For example, the second to fifth paragraph in the discussion should rather be at the end of the manuscript as they provide some conclusions based on the study findings. I would suggest to structure the discussion as follows: 1.) Short summary of the main study aim with one sentence about the innovativeness of the present study, 2.) half a page summary of the main study findings (without presenting precise numbers), 3.) more in-depth discussion of the study findings in comparison to previous research, 4.) Study limitation, 5.) Future studies, 6.) Main conclusions.

Modified as requested.

---

## [Decision Letter · Decision Letter 1]

24 Feb 2021

PONE-D-20-32711R1

Sexual Inhibition and Sexual Excitation in a Sample of Polish Women.

PLOS ONE

Dear Dr. Nowosielski,

Thank you for submitting your manuscript to PLOS ONE. After careful consideration, we feel that it has merit but does not fully meet PLOS ONE’s publication criteria as it currently stands. Therefore, we invite you to submit a revised version of the manuscript that addresses the points raised during the review process.

We look forward to receiving your revised manuscript.

Kind regards,

Julia Velten

Academic Editor

PLOS ONE

Journal Requirements:

Additional Editor Comments (if provided):

Thanks for the revision of your paper.

Comments by the Editor

The manuscript has been revised by an English speaker, however, there are still some typos and mistakes (e.g., sentence structure, line 57, “self-report questionnaire”, line 88, Arousal Contingency (not Contingence), SESII-WW, line 103, refused, line 201) left. Please include another round of language revision before resubmission.

Line 62: Please exchange the dated term “sexual drive” with sexual desire or sexual motivation.

Line 155: Central/Eastern European

Line 179: Please indicate the expected direction of the relationships

Line 289: Needle sharing is not considered a sexual behavior per se

Line 307-309: Aggregating? Also, please reformat this statement.

Line 338: The version of the R software is not the same as the packages (e.g., lavaan, psych, semtools) used for the analysis. Please include the packages used in the description.

Line 372: SISII-W-PL?

Line 477: removed excluded

Line 503: I assume that you used the Chi²-test to determine measurement invariance. However, the differences in CFI and RMSEA seem to suggest that the model is measurement invariant across education groups. Please consider using changes in these values as a basis for determining invariance.

Line 578: the discussion is quite lengthy and I suggest condensing it to a max. of seven pages to improve readability.

Reviewers' comments:

Reviewer's Responses to Questions

**Comments to the Author**

1. If the authors have adequately addressed your comments raised in a previous round of review and you feel that this manuscript is now acceptable for publication, you may indicate that here to bypass the “Comments to the Author” section, enter your conflict of interest statement in the “Confidential to Editor” section, and submit your "Accept" recommendation.

Reviewer #1: (No Response)

2. Is the manuscript technically sound, and do the data support the conclusions?

Reviewer #1: Yes

3. Has the statistical analysis been performed appropriately and rigorously? 

Reviewer #1: Yes

4. Have the authors made all data underlying the findings in their manuscript fully available?

Reviewer #1: Yes

5. Is the manuscript presented in an intelligible fashion and written in standard English?

Reviewer #1: Yes

6. Review Comments to the Author

Reviewer #1: Thank you for addressing most of the comments on my previous review, and improving the overall written style of the manuscript. Overall this seems to be a thorough validation study and will make a useful contribution to the literature. My remaining comments are minor.

1) For the third sentence of the introduction, instead of "they moderate mutual activities" it might be more clear to say "they mutually moderate behavior".

2) For the second paragraph of the introduction, can a citation be provided for Gray's work on general inhibition and excitation?

3) For the rationale and hypotheses of the current study, I still think that it would better to discuss some of the material brought up in the discussion (e.g., the consideration of the impact of Catholic and post-Soviet sociocultural contexts) in the introduction instead. This is because 1) you are making a case that the East/Central European context is important to consider despite previous studies in primarily West European countries, and 2) you are specifically hypothesizing differences due to sociocultural factors. Therefore, it may be best to provide some context for these sociocultural differences when they are brought up in the introduction. And it would allow you to save some space in the discussion as well since you won't need to go over the information in detail there anymore.

4) For the Cronbach's alphas of other measures in the current study, I still think that it would be more useful to report the numbers for the current data. Or, at least perhaps a direct statement saying that all alphas were comparable to the reported literature values would be useful.

5) For the discussion of association of lower scores on both SE & SI with age in the current sample, did the authors consider a possible response bias or some other cohort effect that might differentially impact the reporting, awareness, or participation of younger versus older Polish women?

6) For the sentence that begins on line 749, it refers to "psychotherapy with 5-HTPA1 receptor agonist...", should this be pharmacotherapy? As well, for this part of the discussion, would psychotherapy for anxiety about sexuality be indicated for worries about imperfect sexual circumstances?

7) For the paragraph starting on 765, could the authors clarify how the SESII-W scale might be used in order to facilitate education, reduce anxiety, etc.?

7. PLOS authors have the option to publish the peer review history of their article (what does this mean?). If published, this will include your full peer review and any attached files.

Reviewer #1: No

---

## [Author Response · Author response to Decision Letter 1]

16 Mar 2021

Comments by the Editor

Dear Editor,

Thank you for your valuable remarks. The answers are written below.

1. The manuscript has been revised by an English speaker, however, there are still some typos and mistakes (e.g., sentence structure, line 57, “self-report questionnaire”, line 88, Arousal Contingency (not Contingence), SESII-WW, line 103, refused, line 201) left. Please include another round of language revision before resubmission.

Corrected, as requested

2. Line 62: Please exchange the dated term “sexual drive” with sexual desire or sexual motivation.

Corrected, as requested

3. Line 155: Central/Eastern European

Corrected, as requested

4. Line 179: Please indicate the expected direction of the relationships

Added, as requested

5. Line 289: Needle sharing is not considered a sexual behavior per se

That was change to “or drug injection with shared needles leading to sex”

6. Line 307-309: Aggregating? Also, please reformat this statement.

Corrected, as requested

7. Line 338: The version of the R software is not the same as the packages (e.g., lavaan, psych, semtools) used for the analysis. Please include the packages used in the description.

Added, as requested

8. Line 372: SISII-W-PL?

That abraviation indicates the Polish version of the scale

Explained in the text, as requested.

9. Line 477: removed excluded

Corrected, as requested

10. Line 503: I assume that you used the Chi²-test to determine measurement invariance. However, the differences in CFI and RMSEA seem to suggest that the model is measurement invariant across education groups. Please consider using changes in these values as a basis for determining invariance.

Cleared, as requested.

11. Line 578: the discussion is quite lengthy and I suggest condensing it to a max. of seven pages to improve readability.

That you for that suggestion. Some changes was made, as suggested. However, we think that further condensing the discussion would affect the quality of this detailed analysis of obtained results. Now the Discussion section has six pages plus 1.5 for Clinical Implication, one for Limitation and 0.5 for Conclusions. We hope that now it will meet your expectations. 

Reviewer #1: Thank you for addressing most of the comments on my previous review, and improving the overall written style of the manuscript. Overall this seems to be a thorough validation study and will make a useful contribution to the literature. My remaining comments are minor.

Dear Reviewer,

Thank you for your valuable comments. The answer are written below.

1) For the third sentence of the introduction, instead of "they moderate mutual activities" it might be more clear to say "they mutually moderate behavior".

Corrected, as requested.

2) For the second paragraph of the introduction, can a citation be provided for Gray's work on general inhibition and excitation?

Added, as suggested

3) For the rationale and hypotheses of the current study, I still think that it would better to discuss some of the material brought up in the discussion (e.g., the consideration of the impact of Catholic and post-Soviet sociocultural contexts) in the introduction instead. This is because 1) you are making a case that the East/Central European context is important to consider despite previous studies in primarily West European countries, and 2) you are specifically hypothesizing differences due to sociocultural factors. Therefore, it may be best to provide some context for these sociocultural differences when they are brought up in the introduction. And it would allow you to save some space in the discussion as well since you won't need to go over the information in detail there anymore.

Modified as suggested.

4) For the Cronbach's alphas of other measures in the current study, I still think that it would be more useful to report the numbers for the current data. Or, at least perhaps a direct statement saying that all alphas were comparable to the reported literature values would be useful.

All Cronbach's alphas for the current data are presented in Table 5

5) For the discussion of association of lower scores on both SE & SI with age in the current sample, did the authors consider a possible response bias or some other cohort effect that might differentially impact the reporting, awareness, or participation of younger versus older Polish women?

Yes, we analyzed that in context of age and social desirability. However, we have not found any differences between different age-groups (especially younger vs older). It might be speculated that if the age range in the current study was wider (e.g. up to 80 years old) the association could be different. But that need further studies. Social desirability did not have any influence either. 

6) For the sentence that begins on line 749, it refers to "psychotherapy with 5-HTPA1 receptor agonist...", should this be pharmacotherapy? As well, for this part of the discussion, would psychotherapy for anxiety about sexuality be indicated for worries about imperfect sexual circumstances?

Is should be: Prior research suggests that psychotherapy combined with 5-hydroxytryptamine1A receptor agonist plus Phosphodiesterase-5 inhibitors (PDE-5i) could be a potential therapeutic option [31] for women with high Arousal Continence (i.e., worries that circumstances of sexual activity will not be “perfect”) and high Concerns about Sexual Function (i.e., performance anxiety). That sentence was changed accordingly. 

7) For the paragraph starting on 765, could the authors clarify how the SESII-W scale might be used in order to facilitate education, reduce anxiety, etc.?

That sentences were changed to: . Filling in the scale may therefore have an educational function – while reading the questions/statements women may learn about sexual responses, possible reasons for performance anxiety, and may acknowledge and accept changes in sexual responses capacity in different life stage. The SE scale, in the same mechanism, may also motivate women to seek sexually exciting situations and individual experiences to increase sexual responses in different models, not only the linear models [16], and irrespective of potential limitations (e.g. physical limitations)

---

## [Editor Report · Decision Letter 2]

22 Mar 2021

Sexual Inhibition and Sexual Excitation in a Sample of Polish Women.

PONE-D-20-32711R2

Dear Dr. Nowosielski,

We’re pleased to inform you that your manuscript has been judged scientifically suitable for publication and will be formally accepted for publication once it meets all outstanding technical requirements.

Kind regards,

Julia Velten

Academic Editor

PLOS ONE
---

## [Editor Report · Acceptance letter]

23 Mar 2021

PONE-D-20-32711R2 

Sexual Inhibition and Sexual Excitation in a Sample of Polish Women. 

Dear Dr. Nowosielski:

I'm pleased to inform you that your manuscript has been deemed suitable for publication in PLOS ONE. Congratulations! Your manuscript is now with our production department. 

Kind regards, 

on behalf of

Dr. Julia Velten 

Academic Editor

PLOS ONE